# Pervasive translation in *Mycobacterium tuberculosis*

Carol Smith[1†], Jill G Canestrari[1†], Archer J Wang[1†], Matthew M Champion[2], Keith M Derbyshire[1,3]*, Todd A Gray[1,3]*, Joseph T Wade[1,3]*

[1]Wadsworth Center, Division of Genetics, New York State Department of Health, Albany, United States; [2]Department of Chemistry and Biochemistry, University of Notre Dame, Notre Dame, United States; [3]Department of Biomedical Sciences, School of Public Health, University at Albany, New York, United States

**Abstract** Most bacterial ORFs are identified by automated prediction algorithms. However, these algorithms often fail to identify ORFs lacking canonical features such as a length of >50 codons or the presence of an upstream Shine-Dalgarno sequence. Here, we use ribosome profiling approaches to identify actively translated ORFs in *Mycobacterium tuberculosis*. Most of the ORFs we identify have not been previously described, indicating that the *M. tuberculosis* transcriptome is pervasively translated. The newly described ORFs are predominantly short, with many encoding proteins of ≤50 amino acids. Codon usage of the newly discovered ORFs suggests that most have not been subject to purifying selection, and hence are unlikely to contribute to cell fitness. Nevertheless, we identify 90 new ORFs (median length of 52 codons) that bear the hallmarks of purifying selection. Thus, our data suggest that pervasive translation of short ORFs in *Mycobacterium tuberculosis* serves as a rich source for the evolution of new functional proteins.

**\*For correspondence:**
keith.derbyshire@health.ny.gov (KMD);
todd.gray@health.ny.gov (TAG);
joseph.wade@health.ny.gov (JTW)

[†]These authors contributed equally to this work

## Editor's evaluation

The use of ribosome profiling in this study allowed for the identification of translated regions of the *Mycobacterium tuberculosis* genome, identifying new genomic regions that undergo active translation. A select set of these appears to have been the subject of purifying evolutionary selection, suggesting that this pervasive translation of short genetic regions serves as the basis for the evolution of new proteins/protein functions.

## Introduction

The canonical mode of bacterial translation initiation begins with the association of a 30 S ribosomal subunit, initiator tRNA, and initiation factors, with the ribosome binding site of an mRNA (*Laursen et al., 2005*). Binding of the 30 S initiation complex to the mRNA involves base-pairing interactions between the mRNA Shine-Dalgarno (S-D) sequence, located a short distance upstream of the start codon, and the anti-S-D sequence in the 16 S ribosomal RNA (rRNA). Local mRNA secondary structure around the ribosome binding site can reduce interaction with the 30 S initiation complex. Translation initiates at a start codon, typically an AUG; less frequently, translation initiation occurs at GUG or UUG, and in rare instances at AUC, AUU, and AUA start codons (*Gvozdjak and Samanta, 2020*; *Hecht et al., 2017*). Hence, the likelihood of translation initiation at a given sequence will depend on the sequence upstream of the start codon, the degree of secondary structure in the region surrounding the start codon, and start codon identity.

Due to the requirement for a 5' untranslated region that includes the S-D sequence, mRNAs translated using the canonical mechanism are referred to as 'leadered'. By contrast, 'leaderless' translation

**eLife digest** How can you predict which proteins an organism can make? To answer this question, scientists often use computer programs that can scan the genetic information of a species for open reading frames – a type of DNA sequence that codes for a protein. However, very short genes and overlapping genes are often missed through these searches.

Mycobacteria are a group of bacteria that includes the species *Mycobacterium tuberculosis*, which causes tuberculosis. Previous work has predicted several thousand open reading frames for *M. tuberculosis*, but Smith et al. decided to use a different approach to determine whether there could be more. They focused on ribosomes, the cellular structures that assemble a specific protein by reading the instructions provided by the corresponding gene.

Examining the sections of genetic code that ribosomes were processing in *M. tuberculosis* uncovered hundreds of new open reading frames, most of which carried the instructions to make very short proteins. A closer look suggested that only 90 of these proteins were likely to have a useful role in the life of the bacteria, which could open new doors in tuberculosis research. The rest of the sequences showed no evidence of having evolved a useful job, yet they were still manufactured by the mycobacteria. This pervasive production could play a role in helping the bacteria adapt to quickly changing environments by evolving new, functional proteins.

initiation occurs on mRNAs that lack a 5' UTR, such that the transcription start site (TSS) and translation start codon coincide. The mechanism of leaderless translation initiation is poorly understood. Until recently, there were few known examples of leaderless mRNAs; leaderless translation in the model bacterium *Escherichia coli* was shown to be rare and inefficient (*Moll et al., 2002*; *Romero et al., 2014*; *Shell et al., 2015*). However, recent studies indicate that leaderless translation initiation is a prevalent and robust mechanism in many bacterial and archaeal species (*Beck and Moll, 2018*). We and others showed that ~25% of all mRNAs in *Mycobacterium smegmatis* and *Mycobacterium tuberculosis* (Mtb) are leaderless (*Cortes et al., 2013*; *Shell et al., 2015*). Moreover, our data suggested that any RNA with a 5' AUG or GUG will be efficiently translated using the leaderless mechanism in *M. smegmatis* (*Shell et al., 2015*).

Bacterial open reading frames (ORFs) are typically identified from genome sequences using automated prediction algorithms (*Besemer and Borodovsky, 2005*; *Delcher et al., 2007*; *Hyatt et al., 2010*). Among the criteria used by these algorithms are ORF length, and the presence of a S-D sequence. Hence, they often fail to identify non-canonical ORFs, including overlapping ORFs (*Burge and Karlin, 1998*), leaderless ORFs (*Beck and Moll, 2018*; *Lomsadze et al., 2018*), and short ORFs (sORFs; encoding small proteins of 50 or fewer amino acids; most algorithms have a lower size limit of 50 codons). Recent studies have revealed hundreds of sORFs in diverse bacterial species (*Orr et al., 2020*; *Sberro et al., 2019*; *Storz et al., 2014*; *Stringer et al., 2021*; *VanOrsdel et al., 2018*; *Weaver et al., 2019*). Some sORFs encode functional small proteins that contribute to cell fitness, whereas other sORFs function as *cis*-acting regulators. In eukaryotes, there have been reports of 'pervasive translation' of thousands of unannotated sORFs, likely due to the imperfect specificity of the translation machinery (*Ingolia et al., 2014*; *Ruiz-Orera et al., 2018*; *Wacholder et al., 2021*). The function, if any, of most of these sORFs and/or their encoded proteins is unclear, although they are rarely subject to purifying selection (*Ruiz-Orera et al., 2018*; *Wacholder et al., 2021*). Nonetheless, a high-throughput mutagenesis study of unannotated sORFs in human cells suggested that some contribute to cell fitness (*Chen et al., 2020*). Moreover, pervasively translated eukaryotic sORFs may function as 'proto-genes', that, over the course of evolution, can acquire a function promoting cell fitness, a process referred to as 'de novo gene birth' (*Blevins et al., 2021*; *Carvunis et al., 2012*; *Ruiz-Orera et al., 2018*; *Vakirlis et al., 2018*; *Vakirlis et al., 2020*).

Ribosome profiling (Ribo-seq) is a powerful experimental approach to identify the translated regions of mRNAs by mapping ribosome-protected RNA fragments (*Ingolia et al., 2009*). Ribo-RET is a modified form of Ribo-seq in which bacterial cells are treated with the antibiotic retapamulin before lysis; retapamulin traps bacterial ribosomes at sites of translation initiation, whereas elongating ribosomes are free to complete translation (*Meydan et al., 2019*). Thus, Ribo-RET facilitates the identification of overlapping ORFs by limiting the signal to the start codons (*Meydan et al., 2018*; *Meydan et al.,*

*2019*). Ribo-RET was recently applied to *E. coli*, revealing start codons for many previously unde-scribed ORFs (*Meydan et al., 2019*; *Stringer et al., 2021*; *Weaver et al., 2019*), including sORFs, and ORFs positioned in frame with annotated ORFs, such that the translated protein is an isoform of the previously described protein. Here, we use a combination of Ribo-seq and Ribo-RET to map trans-lated ORFs in *Mtb*. We detect thousands of robustly translated, previously undescribed sORFs from leaderless and leadered mRNAs. We also identify hundreds of ORFs that have start codons upstream or downstream of those for annotated genes, in the same reading frame. We conclude that the *Mtb* transcriptome is pervasively translated, with spurious translation initiation occurring at many sites. We also identify a subset of novel sORFs that appear to be under purifying selection, suggesting these ORFs, or the proteins they encode, contribute to cell fitness. Thus, our data suggest that pervasive translation of sORFs in *Mtb* serves as a rich source for the evolution of functional genes.

## Results

### Hundreds of actively translated sORFs from leaderless mRNAs

*Mtb* has a genome of 4,411,532bp, with 3989 annotated protein-coding genes (RefSeq annotation). Two previous studies of *Mtb* identified 1285 transcription start sites (TSSs) for which the associated transcript begins with the sequence 'RUG' (R = A or G; *Supplementary file 1A*; *Cortes et al., 2013*; *Shell et al., 2015*), suggesting that these transcripts correspond to leaderless mRNAs (*Shell et al., 2015*). Of the 1285 TSSs associated with a 5' RUG, 577 match the start codons of protein-coding genes included in the current genome annotation, as previously noted (*Cortes et al., 2013*; *Shell et al., 2015*). A further 338 of the RUG-associated TSSs correspond to putative ORFs whose start codons are unannotated, but whose stop codons match those of annotated genes; we refer to this architecture as 'isoform', since translation of these putative ORFs would generate N-terminally extended or trun-cated isoforms of annotated proteins. We note that some isoform ORFs likely reflect mis-annotations, as has been suggested previously (*Cortes et al., 2013*; *Shell et al., 2015*). Lastly, 370 of the 1,285 RUG-associated TSSs correspond to putative ORFs whose start and stop codons do not match those of any annotated gene; we refer to these as putative 'novel' ORFs.

To determine whether the putative isoform and novel leaderless ORFs are actively translated, we performed Ribo-seq in *Mtb*. Note that all genome-scale data described in this manuscript can be viewed in our interactive genome browser (https://mtb.wadsworth.org/). We first assessed ribosome occupancy profiles for leadered ORFs that are present in the current genome annotation. Consistent with previous studies (*Oh et al., 2011*; *Woolstenhulme et al., 2015*), we observed enrichment of ribosome occupancy at start and stop codons of annotated, leadered ORFs; the 3' ends of ribosome-protected RNA fragments are enriched 15 nt downstream of the start codons, and 12 nt downstream of stop codons (*Figure 1A*). We note that there are also smaller peaks and troughs of Ribo-seq signal precisely at start and stop codons, likely attributable to sequence biases associated with library prepa-ration that are highlighted when groups of similar sequences (e.g. start/stop codons) are aligned (see Methods). We next assessed ribosome occupancy profiles for the 577 leaderless ORFs that are present in the current genome annotation. As expected, we observed an enrichment of ribosome-protected RNA fragments, with 3' ends positioned 12 nt downstream of stop codons (*Figure 1B*), consistent with the profile observed for leadered ORFs. However, 3' ends of ribosome-protected RNA fragments were not enriched 15 nt downstream of the start codons of the 577 annotated leaderless ORFs; rather, we observed enrichment spread across the region ~25–35 nt downstream of leaderless start codons (*Figure 1B*), suggesting either that ribosomes at leaderless ORF start codons behave differently to those at leadered ORF start codons, or that ribosome-protected fragments are too small to be represented in the RNA library; this observation is consistent with a previous study (*Sawyer et al., 2021*). Further confounding analysis of leaderless start codons, which are, by definition, aligned with TSSs, we consistently observed non-random Ribo-seq signals at TSSs of non-leaderless tran-scripts (*Figure 1—figure supplement 1*), albeit to a lesser extent than that observed for leaderless gene starts.

We reasoned that if the putative leaderless isoform and novel ORFs are actively translated, they would exhibit similar ribosome occupancy profiles to the leaderless annotated ORFs. Indeed, this was the case, with similar relative occupancy of ribosomes undergoing translation initiation and termina-tion at start/stop codons (*Figure 1C–D*; we did not analyze isoform ORF stop codons because they

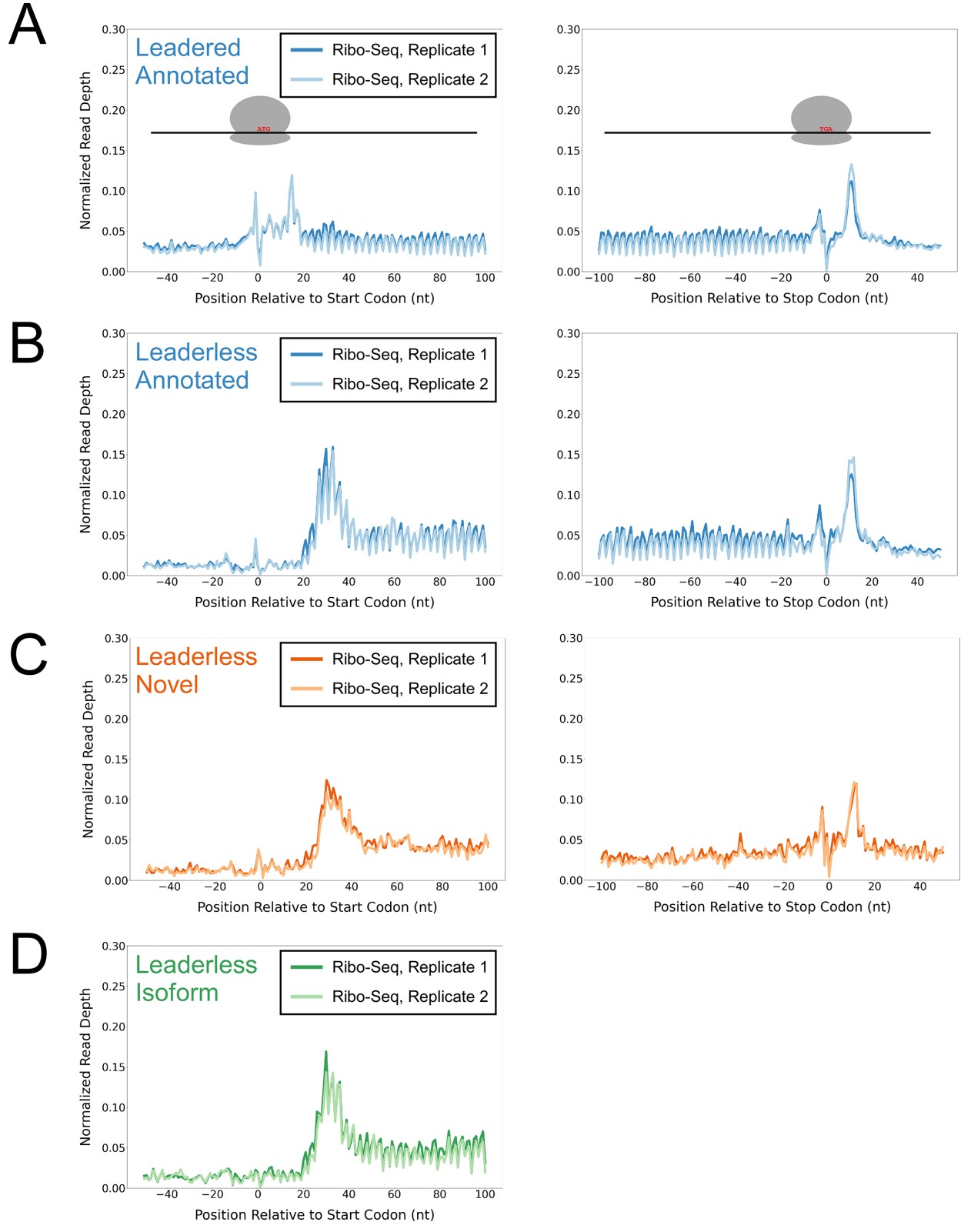

**Figure 1.** Ribo-seq data support the translation of hundreds of isoform and novel ORFs from leaderless mRNAs. (**A**) Metagene plot showing normalized Ribo-seq sequence read coverage for untreated cells in the regions around start (left graph) and stop codons (right graph) of previously annotated, leadered ORFs. Note that sequence read coverage is plotted only for the 3' ends of reads, since these are consistently positioned relative to the ribosome P-site (**Woolstenhulme et al., 2015**). Data are shown for two biological replicate experiments. The schematics show the position of initiating/

*Figure 1 continued on next page*

*Figure 1 continued*

terminating ribosomes, highlighting the expected site of ribosome occupancy enrichment at the downstream edge of the ribosome. (**B**) Equivalent data to (**A**) but for putative annotated, leaderless ORFs. (**C**) Equivalent data to (**A**) but for putative novel, leaderless ORFs. (**D**) Equivalent data to (**A**) but for putative isoform, leaderless ORFs. Only data for start codons are shown because the same stop codon is used by both an annotated and isoform ORF.

The online version of this article includes the following figure supplement(s) for figure 1:

**Figure supplement 1.** Modest enrichment of Ribo-seq signal downstream of the transcription start sites (TSSs) of non-leaderless RNAs.

are shared with those of annotated ORFs). Thus, our data are consistent with active translation of the majority of the 370 putative novel ORFs as leaderless mRNAs. Strikingly, 268 of the leaderless novel ORFs are sORFs. We conclude that *Mtb* has hundreds of actively translated sORFs on leaderless mRNAs.

## Ribo-RET identifies sites of translation initiation in *Mtb*

While there are likely >1000 leaderless mRNAs in *Mtb*, most mRNAs are leadered (*Cortes et al., 2013*; *Sawyer et al., 2021*; *Shell et al., 2015*). Given that our data support the existence of >300 novel ORFs translated from the 5′ ends of leaderless mRNAs, we speculated that there are many more unannotated ORFs translated from leadered initiation codons. While sites of leaderless translation initiation can be readily identified from TSS maps, identification of novel leadered ORFs is more challenging. Translated leadered ORFs generate signal in Ribo-seq datasets, but identification of novel ORFs from Ribo-seq data is confounded by (i) the potential for artifactual signal in 5′ UTRs due to the binding of RNA-binding proteins (*Ji et al., 2016*), and (ii) masking of signal by overlapping ORFs on the same strand. To circumvent these problems, we performed Ribo-RET with *Mtb* to specifically map sites of translation initiation. We aligned the ribosome-protected RNA fragment sequences to the *Mtb* genome to identify 'Initiation-Enriched Ribosome Footprints' (IERFs), sites of ribosome occupancy that exceed the local background (*Supplementary file 1B*). Specifically, IERFs correspond to genomic coordinates that have ribosome occupancy coverage that exceeds an arbitrarily defined threshold value (5.5 reads per million) and is at least 10-fold higher than the mean ribosome occupancy coverage in the region 50 nt upstream to 50 nt downstream. We hypothesized that most IERFs correspond to sites of translation initiation. In support of this idea, there is a strong enrichment of IERF 3′ ends 15 nt downstream of the start codons of annotated, leadered genes; this enrichment is substantially greater than that observed for Ribo-seq data from cells grown without retapamulin treatment (*Figure 2A*; *Figure 2—figure supplement 1*).

We determined the abundance of all trinucleotide sequences in the 40 nt regions upstream of IERF 3′ ends; there is a > 2 fold enrichment of ATG, GTG and TTG (likely start codons), but not CTG, ATT or ATC, 15 nt upstream of IERF 3′ ends, and an enrichment of AGG and GGA (components of a consensus AGGAGGU Shine-Dalgarno sequence) 22–31 nt upstream of IERF 3′ ends (*Figure 2B*). We also observed >1.5 fold enrichment of ATG and GTG 14, 16, 17, and 18 nt upstream of IERF 3′ ends. The enrichment and position of start codon and Shine-Dalgarno-like sequence features upstream of IERFs are consistent with IERFs marking sites of translation initiation. We observed a strong enrichment of A/T immediately 3′ of the IERFs, i.e. on the other side of the site cleaved by micrococcal nuclease (MNase) during the Ribo-RET procedure; 'A' was found most frequently (53% of IERFs), and 'G' found the least frequently (2% of IERFs; *Figure 2—figure supplement 2*). This sequence bias is likely not due to a biological phenomenon, but rather to the sequence preference of MNase, which is known to display sequence bias when cutting DNA (*Dingwall et al., 1981*) and RNA (*Woolstenhulme et al., 2015*). The sequence bias is apparent in the complete Ribo-RET libraries, with 74% of sequenced ribosome-protected fragments having an 'A' or 'U' 3′ of the upstream MNase site. Given that the genomic A/T content in *Mtb* is only 34%, it is likely that inefficient RNA processing by MNase led to an underrepresentation of some G/C-rich translation initiation sites in the Ribo-RET data, and may explain the extended footprints ( > 15 nt) in G/C-rich contexts (see Discussion). This sequence bias also likely favors cleavage precisely at exposed start codons, which are strongly enriched for A/T bases, creating more RNA library fragments that end in these sequences (e.g. enriched Ribo-seq signal precisely at start codons in *Figure 2A*).

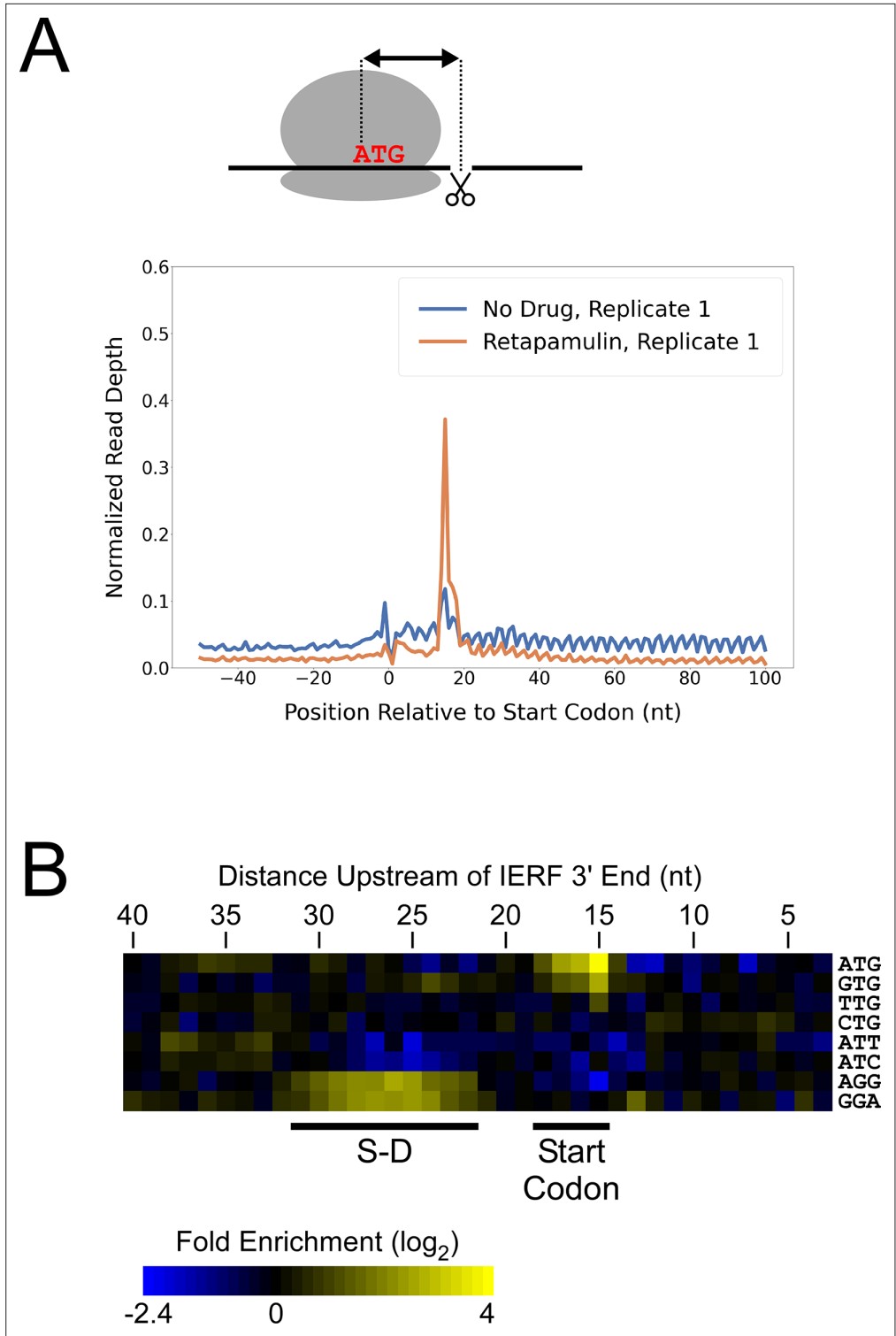

**Figure 2.** Ribo-RET of *M. tuberculosis* identifies sites of translation initiation. (**A**) Metagene plot showing normalized Ribo-seq and Ribo-RET sequence read coverage (single replicate for each; data indicate the position of ribosome footprint 3' ends) in the region from –50 to +100 nt relative to the start codons of annotated, leadered ORFs. (**B**) Heatmap showing the enrichment of eight selected trinucleotide sequences, for regions upstream of IERFs, relative to control regions. Expected positions of start codons and S-D sequences are indicated below the heatmap.

The online version of this article includes the following figure supplement(s) for figure 2:

*Figure 2 continued on next page*

*Figure 2 continued*
**Figure supplement 1.** Retapamulin treatment traps initiating ribosomes.
**Figure supplement 2.** Sequence bias associated with the 3' ends of ribosome-protected RNAs at IERFs.

## Identification of putative ORFs from Ribo-RET data

A total of 1994 IERFs were found in both replicate experiments (*Supplementary file 1B*). 71% (1406) of these IERFs were associated with a potential ATG or GTG start codon 14–18 nt upstream of their 3' ends, or a potential TTG start codon 15 nt upstream of their 3' ends (*Supplementary file 1C*), a far higher proportion than that expected by chance (17%). Thus, these 1,406 IERFs correspond to the start codons of putative ORFs, with an overall estimated false discovery rate (FDR) of 9% (see Materials and methods for details). 34% (478; FDR of 0.3%) of the putative ORFs precisely match previously annotated ORFs; 27% (373; FDR of 9%) overlap , and are in frame with previously annotated ORFs (i.e. isoform ORFs); 39% (555; FDR of 15%) are novel ORFs, with no match to a previously annotated stop codon. A total of 112 novel ORFs were found entirely in regions presently designated as intergenic; the remaining novel ORFs overlap partly or completely with annotated genes in sense and/or antisense orientations (*Figure 3A*; *Supplementary file 1C*). Strikingly, 77% (430) of the novel ORFs we identified are sORFs, with 48 novel ORFs consisting of only a start and stop codon (*Supplementary file 1C*), an architecture recently described in *E. coli* (*Meydan et al., 2019*).

We reasoned that if the isoform ORFs and novel ORFs are genuine, they should have S-D sequences upstream, and their start codons should each be associated with a region of reduced RNA secondary structure, as has been described for ORFs in other bacterial species (*Baez et al., 2019*; *Del Campo et al., 2015*). As we had observed for the set of all IERFs, regions upstream of isoform ORFs and novel ORFs are associated with an enrichment of AGG and GGA sequences in the expected location of a S-D sequence (*Figure 3—figure supplement 1*). This enrichment is lower than for annotated genes, but it is important to note that a S-D sequence was likely a contributing criterion in computationally predicting the initiation codons of annotated genes. We also assessed the level of RNA secondary structure upstream of all the putative ORFs identified by Ribo-RET. The predicted secondary structure for a set of random genomic sequences was significantly higher than the predicted secondary structure around the start of the identified annotated, novel, or isoform ORFs (Mann-Whitney U Test $P <$ $2.2e^{-16}$ in all cases; *Figure 3B*). Moreover, the predicted secondary structure around the start of the annotated ORFs was only modestly, albeit significantly, higher than that of novel ORFs (Mann-Whitney U Test $P = 1e^{-3}$). Collectively, the ORFs identified from Ribo-RET data exhibit the expected features of genuine translation initiation sites.

## ORFs identified by Ribo-RET are actively translated in untreated cells

To determine if isoform ORFs and novel ORFs are actively and fully translated in cells not treated with retapamulin, we analyzed Ribo-seq data generated from cells grown without drug treatment. We assessed ribosome occupancy for annotated, novel, and isoform ORFs identified by Ribo-RET. As for the predicted leaderless ORFs, we reasoned that expressed leadered ORFs would be associated with increased ribosome occupancy at start and stop codons, as exemplified by previously annotated, leadered ORFs (*Figure 1A*; *Oh et al., 2011*; *Woolstenhulme et al., 2015*). Accordingly, annotated ORFs identified by Ribo-RET were strongly enriched for Ribo-seq signal 15 nt downstream of their start codons and 12 nt downstream of their stop codons (*Figure 4A–B*). We observed similar Ribo-seq enrichment profiles at the start and stop codons of novel ORFs, and downstream of the start codons of isoform ORFs (*Figure 4A and C–D*), but we did not observe these enrichment profiles for a set of mock ORFs (*Figure 4—figure supplement 1A*). Moreover, we did not observe enrichment of RNA-seq signal at start/stop codons, ruling out biases associated with library construction (*Figure 4—figure supplement 1B-D*). Overall, our data are consistent with most Ribo-RET-predicted isoform and novel ORFs being actively translated from start to stop codon, independent of retapamulin treatment.

## Identification of lower-confidence ORFs from Ribo-RET data

In addition to the 1994 IERFs present in both replicates of Ribo-RET data, 4216 IERFs were found in only the first replicate dataset, which was associated with a stronger enrichment of ribosome occupancy at start codons (compare *Figure 2A* and *Figure 2—figure supplement 1*). Strikingly, 2791

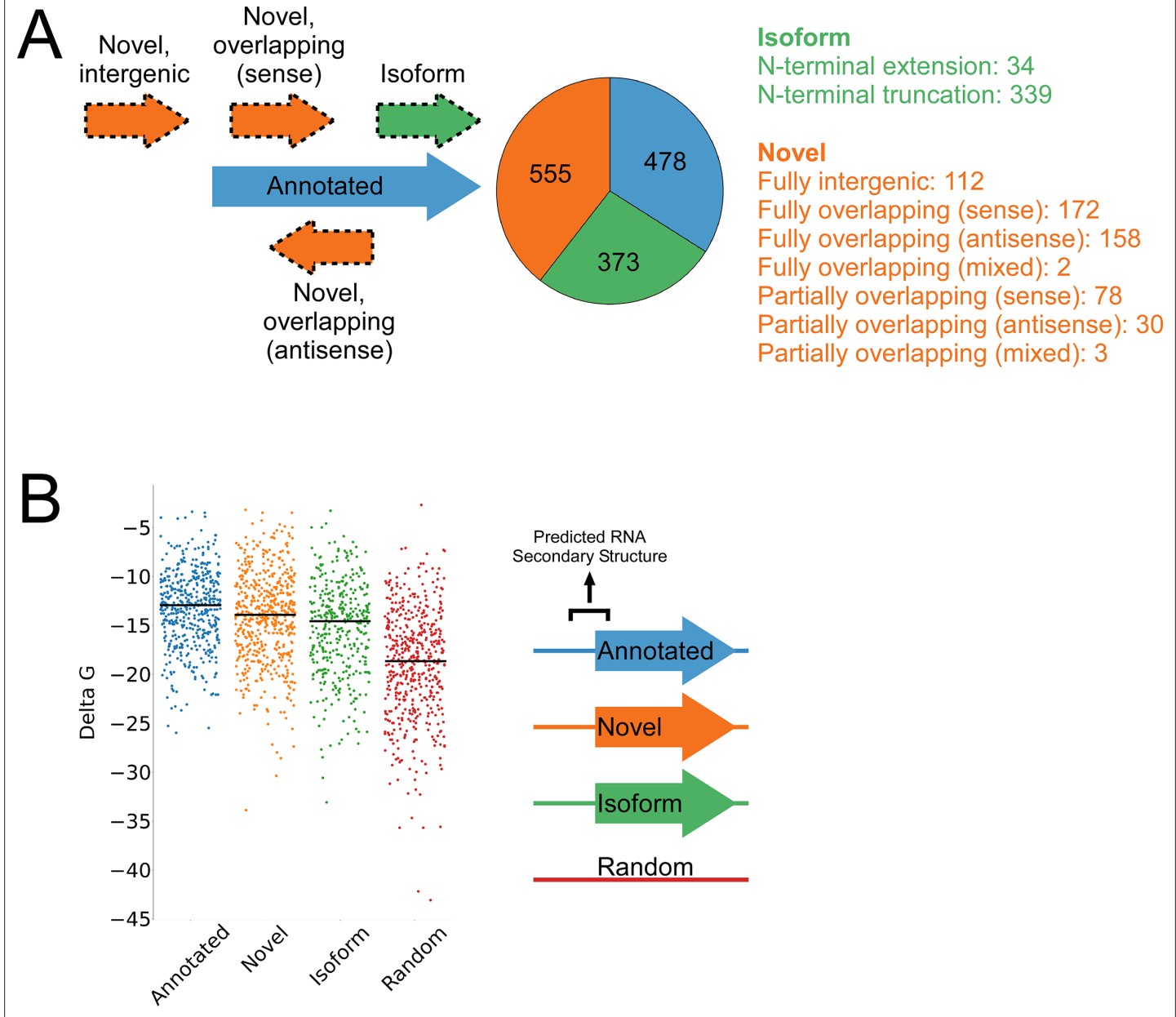

**Figure 3.** Features of higher-confidence ORFs identified by Ribo-RET. (**A**) Distribution of different classes of ORFs identified by Ribo-RET. The pie-chart shows the proportion of identified ORFs in each class. Isoform ORFs are further classified based on whether they are longer ('N-terminal extension') or shorter ('N-terminal truncation') than the corresponding annotated ORF. Novel ORFs are further classified based on their overlap with annotated genes. 'Sense', 'antisense', and 'mixed' refer to whether the overlapping gene(s) is/are in the sense, antisense, or both (multiple overlapping genes) orientations with respect to the novel ORF. 'Fully' and 'Partially' indicate whether all or only some of the novel ORF overlaps annotated genes. (**B**) Strip plot showing the ΔG for the predicted minimum free energy structures for the regions from –40 to +20 nt relative to putative start codons for the different classes of ORF, and for a set of 500 random sequences. Median values are indicated by horizontal lines.

The online version of this article includes the following figure supplement(s) for figure 3:

**Figure supplement 1.** Enrichment of SD-like sequences upstream of higher-confidence ORFs identified by Ribo-RET.

(66%) of IERFs found in only the first Ribo-RET dataset were associated with a potential start codon 14–18 nt upstream of their 3' ends (*Supplementary file 1C*; see Materials and methods for details), a far higher proportion than that expected by chance (17%), and a similar proportion to that observed for IERFs found in both replicate Ribo-RET datasets (70%). We refer to ORFs identified from only the first Ribo-RET dataset as 'lower-confidence' ORFs, reflecting the marginally higher FDRs; we refer to

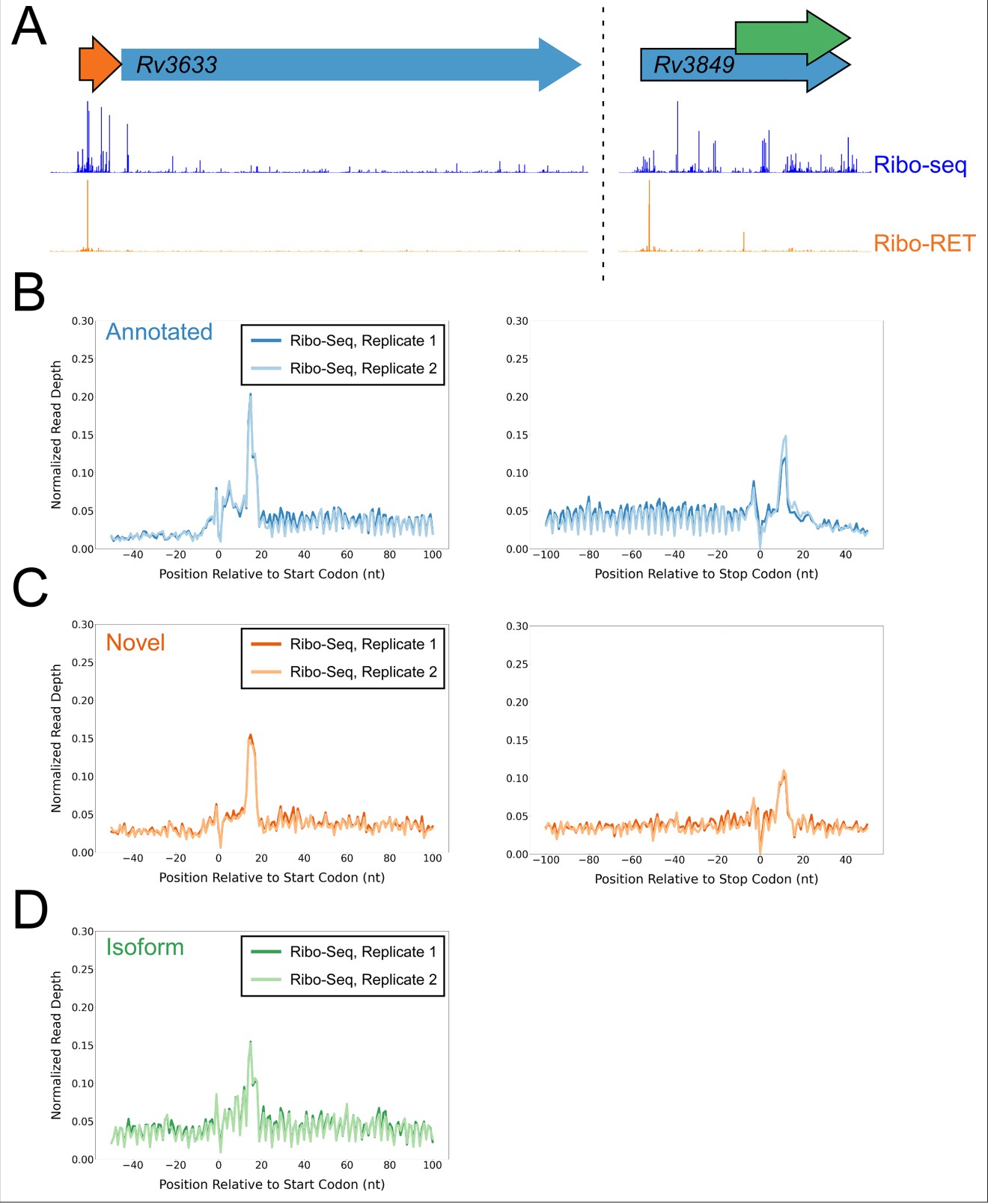

**Figure 4.** Ribo-seq data support the translation of hundreds of isoform and novel ORFs identified by Ribo-RET. (**A**) Ribo-seq and Ribo-RET sequence read coverage (read 3′ ends) across two genomic regions, showing examples of putative ORFs in the annotated (blue arrow), novel (orange arrow), and isoform (green arrow) categories. ORFs identified by Ribo-RET shown with a black outline. (**B**) Metagene plot showing normalized Ribo-seq sequence read coverage (data indicate the position of ribosome footprint 3′ ends) for untreated cells in the regions around start (left graph) and stop codons (right

*Figure 4 continued on next page*

*Figure 4 continued*

graph) of ORFs predicted from Ribo-RET profiles, that correspond to previously annotated genes. (**C**) Equivalent data to (**B**) but for putative novel ORFs identified from Ribo-RET data. (**D**) Equivalent data to (**B**) but for putative isoform ORFs identified from Ribo-RET data. Only data for start codons are shown because the same stop codon is used by both an annotated and isoform ORF.

The online version of this article includes the following figure supplement(s) for figure 4:

**Figure supplement 1.** Control analyses using mock ORFs or RNA-seq data.

**Figure supplement 2.** Features of lower-confidence ORFs identified by Ribo-RET.

ORFs identified from both Ribo-RET datasets as 'higher-confidence' ORFs. 22% (614; FDR of 0.6%) of the lower-confidence ORFs are annotated, 29% (801; FDR of 10%) are isoform, and 49% (1372; FDR of 16%) are novel. 77% (1061) of the novel lower-confidence ORFs are sORFs, with 120 consisting of only a start and stop codon (*Figure 4—figure supplement 2A*), mirroring the proportions observed in the higher-confidence dataset.

Regions upstream of lower-confidence annotated, novel, and isoform ORFs are associated with an enrichment of AGG and GGA sequences in the expected location of a Shine-Dalgarno sequence (*Figure 4—figure supplement 2B*). The predicted secondary structure for a set of random genomic sequences was significantly higher than the predicted secondary structure around the start of the lower-confidence annotated ORFs, novel ORFs, and isoform ORFs (Mann-Whitney U Test $P < 2.2e^{-16}$ in all cases; *Figure 4—figure supplement 2C*). Moreover, the predicted secondary structure around the start of the lower-confidence annotated ORFs was not significantly higher than that of the lower-confidence novel ORFs (Mann-Whitney U Test $P = 0.22$). Lastly, we examined ribosome occupancy at the start and stop codons of the lower-confidence ORFs from our Ribo-seq data generated from cells grown without drug treatment. Lower-confidence annotated, novel, and isoform ORFs were strongly enriched for Ribo-seq signal 15 nt downstream of their start codons and 12 nt downstream of their stop codons (*Figure 4—figure supplement 2D-F*). Collectively, the lower-confidence ORFs exhibit the characteristics of actively translated regions.

## Novel ORFs tend to be weakly transcribed but efficiently translated

To investigate how efficiently novel ORFs are expressed, we determined RNA levels from RNA-seq data, and ribosome occupancy levels from Ribo-seq data, for all annotated and novel ORFs detected in this study (leaderless and leadered ORFs). We also determined RNA and ribosome occupancy levels for putatively untranslated regions of 1854 control transcripts (see Materials and methods for details). For novel ORFs, we analyzed only the 871 ORFs for which ≥ 50 nt of the ORF is ≥30 nt from an annotated gene on the same strand, to avoid overlapping signal from other ORFs. As a group, novel ORFs have lower RNA levels and lower ribosome occupancy levels than the 1670 annotated ORFs (*Figure 5A* top panel; *Figure 5—figure supplement 1A* top panel; *Figure 5—figure supplement 1B-C*). By contrast, the non-coding control transcripts as a group have similar RNA levels to novel ORFs, but lower ribosome occupancy levels (*Figure 5A*, lower panels; *Figure 5—figure supplement 1A* lower panels; *Figure 5—figure supplement 1B-C*). To estimate the ribosome occupancy per transcript, we determined the ratio of Ribo-seq reads to RNA-seq reads for each region analyzed (*Figure 5B*; *Supplementary file 1*, tabs A + C). As a group, novel ORFs have only slightly lower ribosome occupancy per transcript than annotated ORFs, while both novel and annotated ORFs have markedly higher ribosome occupancy per transcript than the control non-coding transcripts. We conclude that the RNA level for novel ORFs tends to be lower than that for annotated ORFs, but novel ORFs are translated with similar efficiency to annotated ORFs, and are thus clearly distinct from non-coding transcripts. The overall lower expression of novel ORFs relative to annotated ORFs is also reflected by lower Ribo-RET occupancy at their start codons (*Figure 5—figure supplement 2*).

## Validation of novel ORFs using mass spectrometry

Mass spectrometry (MS) provides a rigorous methodology to define the *Mtb* proteome. However, we predict that many of the small proteins we describe here are likely to be missed by MS because (i) there are biases against retaining small proteins in standard sample preparation methods and, (ii) small proteins generate few tryptic peptides. We hypothesized that we could enrich for small proteins by processing the normally discarded fractions from each of two standard preparations (*Wisniewski*

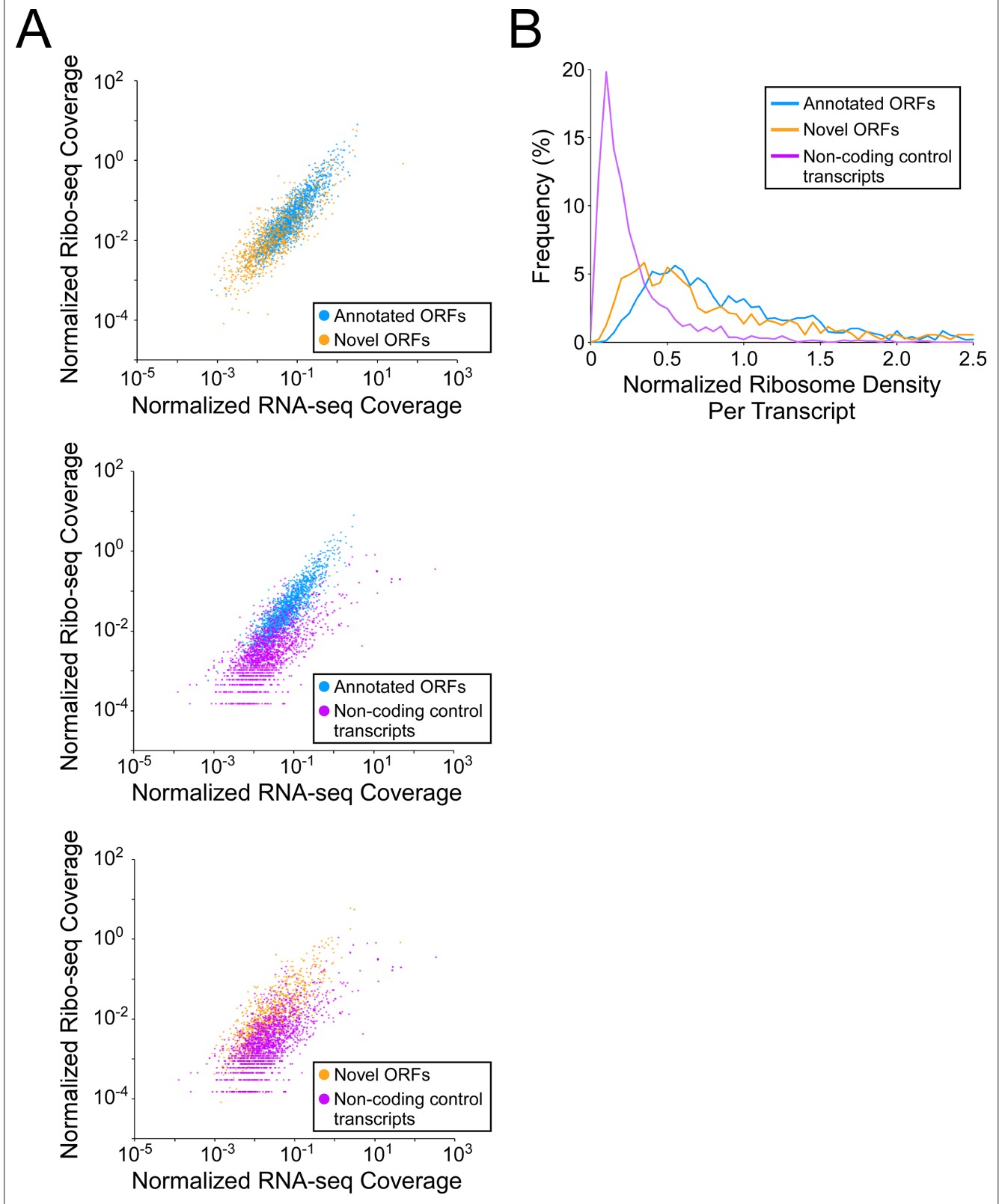

**Figure 5.** Novel ORFs are efficiently translated. (**A**) Pairwise comparison of normalized RNA-seq and Ribo-seq coverage for annotated, novel and non-coding control transcripts. Reads are plotted as RPM per nucleotide using a single replicate of each dataset for reads aligned to the reference genome at their 3' ends. The categories compared are: (**i**) annotated ORFs (higher-confidence and lower-confidence ORFs detected by Ribo-RET, and leaderless ORFs; blue datapoints), (**ii**) novel ORFs (higher-confidence and lower-confidence ORFs detected by Ribo-RET and leaderless ORFs, for regions at least

*Figure 5 continued on next page*

*Figure 5 continued*

30 nt from an annotated gene; orange datapoints), and (iii) a set of 1854 control transcript regions that are expected to be non-coding (see Materials and methods; purple datapoints). ORF/transcript sets are plotted in pairs to aid visualization. (**B**) Normalized ribosome density per transcript (ratio of Ribo-seq coverage to RNA-seq coverage) for the same sets of ORFs/transcripts. The graph shows the frequency (%) of ORFs/transcripts within each group for bins of 0.05 density units.

The online version of this article includes the following figure supplement(s) for figure 5:

**Figure supplement 1.** Novel and isoform ORFs are expressed at lower levels than annotated ORFs.

**Figure supplement 2.** Novel and isoform ORF start codons have lower ribosome occupancy than annotated ORF start codons in Ribo-RET data.

*et al., 2009*). In total, we analyzed five samples prepared in different ways designed to enrich for small proteins (see Materials and methods). We also analyzed a sample made by in-solution digestion, which does not discard small proteins during final preparative stages (see Materials and methods). Nano-UHPLC-MS/MS on these samples identified proteins encoded by 44 of the putative leaderless and leadered novel ORFs identified in this study, at an estimated overall FDR of 1% (*Tang et al., 2008*). Novel proteins detected by MS are indicated in *Supplementary file 1A, C*. Eight proteins were detected in more than one preparation, or with independent peptide matches. Direct analysis from the mixed-organic extraction (with and without demethylation), and analysis of a minimally treated in-solution digestion, yielded the majority of the protein identifications. Ten of the proteins we detected are <50 amino acids in length, with the shortest being 23 amino acids long. The methods aimed at enriching for small proteins detected proteins of a smaller average size: the mean predicted length of novel proteins identified with small protein enrichment strategies was 60 amino acids, versus 86 amino acids for proteins identified from in-solution digestion. We anticipate that additional modifications in the enrichment protocols for small proteins will further improve the sensitivity of detection of small proteins.

Since many small proteins were only identified as single peptides by MS, we sought a direct approach to validate their detection. Three MS-detected novel small proteins were commercially synthesized, and their MS/MS spectra determined for empirical comparison to the native small protein. The three proteins were selected from high- (local FDR < 1%), and medium- (local FDR < 5%) search scores. Two of these proteins are translated from leaderless ORFs and one from a leadered ORF. For all three proteins, the numerical ions from the synthetic peptide matched those from the proteomic datasets, with conservation of the mass intensity (*Figure 6*). We conclude that all three proteins are translated as stable products that match the sequence expected based on Ribo-RET data.

## Validation of novel and isoform start codons using reporter gene fusions

We sought to validate selected novel and isoform ORFs. We hypothesized that the start codons identified by Ribo-RET would direct translation initiation in a reporter system that controls for extraneous contextual variables. We selected 18 novel predicted start codons that scored in the top quartile for ribosome occupancy in Ribo-RET datasets; in Ribo-seq profiles, the associated ribosome densities per transcript cover a broad range of values (median percentile rank of 37 for the eight ORFs that could be assessed). We tested these start codons by fusing them to a luciferase reporter, including 25 bp of upstream sequence for each ORF tested. We constructed equivalent reporter fusions with a single base substitution in the predicted start codon (RTG to RCG). For comparison, we included wild-type and start codon mutant luciferase reporter fusions for three annotated ORFs (*icl1*, *sucC*, and *mmsA*). The reporter plasmids were integrated into the chromosome of *M. smegmatis*. Luciferase expression from each of the 20 luciferase fusions, including those for five novel ORFs from our lower-confidence list, was significantly reduced by mutation of the start codon (*Figure 6—figure supplement 1*; p < 0.05 or 0.01, as indicated, one-way Student's T-test). Mutation of the start codons reduced, but did not abolish, luciferase expression; this was true even for the three annotated ORFs. We speculate that translation can initiate at low levels from non-canonical start codons, as has been described for *E. coli* (*Hecht et al., 2017*). We note that our plasmid reporter system was designed to minimize extraneous variables between constructs that could confound initiation codon evaluation, which necessarily removed the candidate start codons from their larger native context. Overall, the luciferase reporter fusion data are consistent with active translation from the start codons identified by Ribo-RET.

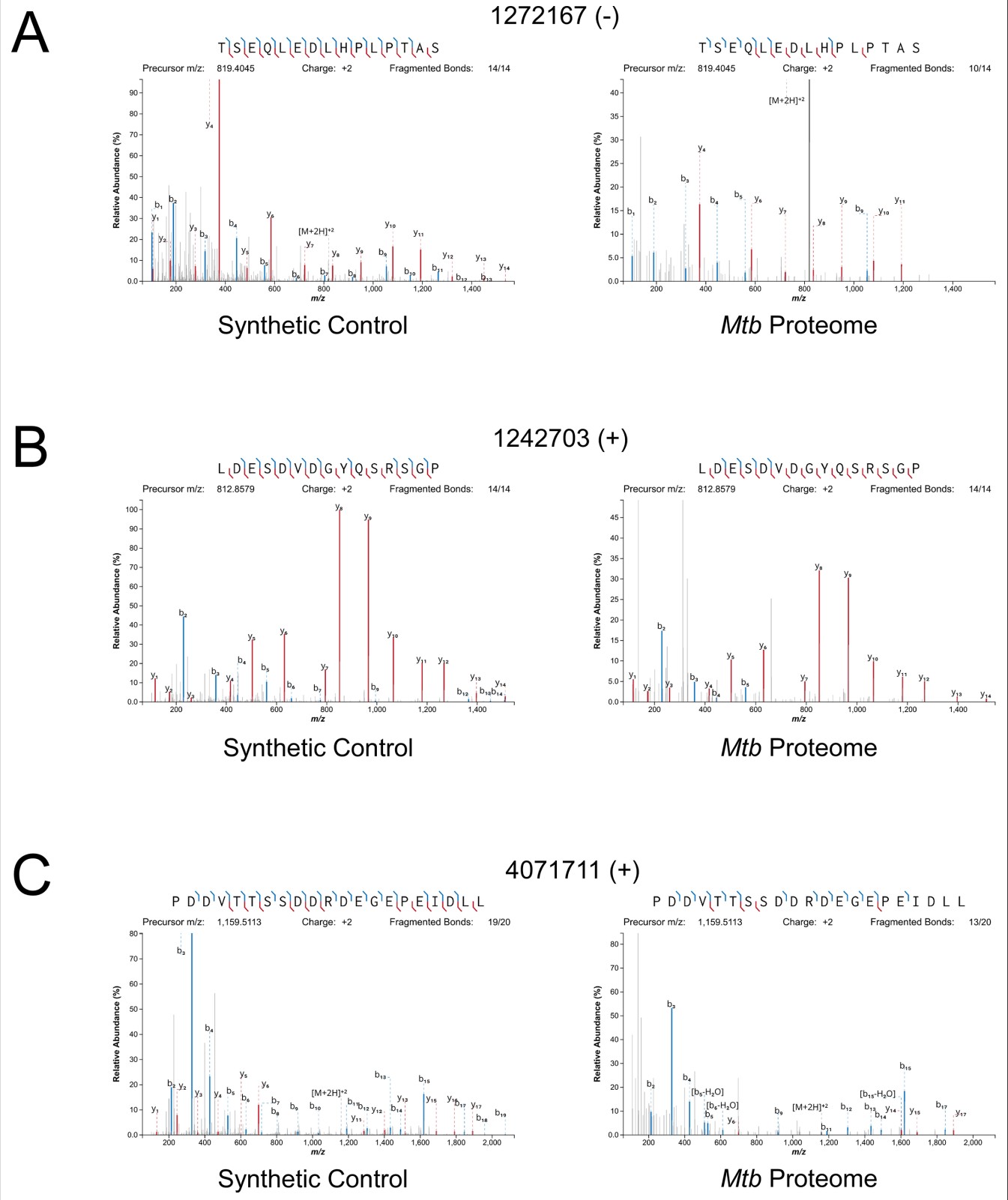

**Figure 6.** Mass spectrometry validation of selected ORFs. MS/MS spectra from novel ORFs measured with a synthetic peptide compared to spectra measured from the *Mtb* proteome. The genome coordinate and strand of each selected novel ORF start codon is indicated. (**A**) Leaderless ORF 1272167 (-) was identified from amino-acids 2–24. The $y_4$ and parent m/z ions are off-scale. (**B**) Leaderless ORF 1242703 (+) was observed from amino acids 46–61. (**C**) Leadered ORF 4071711 (+) was observed from amino acids 4–26. The $b_3$ ion is off-scale. Measured b-ions are in blue, and y-ions are in

*Figure 6 continued on next page*

*Figure 6 continued*

red. The nearly complete spectrum obtained for each peptide and the fragment-mass balance clearly indicate that these sORFs are identical to their synthetic cognates.

The online version of this article includes the following source data and figure supplement(s) for figure 6:

**Figure supplement 1.** Validation of selected novel and isoform ORFs using luciferase reporter fusions.

**Figure supplement 2.** Validation of selected novel and isoform ORFs by western blot.

**Figure supplement 2—source data 1.** Images of full western blots are provided.

## Validation of novel and isoform ORFs using western blotting

To directly assess translation of selected putative ORFs, we generated constructs for two complete novel ORFs with 3 x FLAG tags fused at the encoded C-terminus. We generated equivalent constructs with a single base substitution in the putative start codon. The tagged constructs were integrated into the chromosome of *M. smegmatis*. The two proteins were detected by western blot, and they were not detected from cells with mutant start codons (*Figure 6—figure supplement 2A*). We generated equivalent 3 x FLAG-tagged strains for two isoform ORFs. We detected the overlapping, full-length protein by western blot, and expression of these full-length proteins was unaffected by mutation of the isoform ORF start codon (*Figure 6—figure supplement 2B*). We also detected a protein of smaller size, corresponding to the expected size of the isoform protein; expression of these small isoform proteins was not detected in the start codon mutant constructs (*Figure 6—figure supplement 2B*). Notably, for the pairs of novel and isoform proteins we detected by western blot, the two more highly expressed proteins were from the lower-confidence set of ORFs. Overall, these data support the ORF predictions from the Ribo-RET data, and the existence of novel and isoform ORFs identified from only a single replicate of Ribo-RET data.

## Limited G/C-Skew in the codons of non-overlapping novel ORFs

The *Mtb* genome has a high G/C content (65.6%). There is a G/C bias within codons of annotated genes: the second position of codons is particularly constrained to encode specific amino acids, which supersedes the G/C bias of the genome, whereas the third (wobble) position has few such constraints. Hence, functional ORFs under purifying selection exhibit G/C content below the genome average at the second codon position and above the genome average at the third codon position (*Bibb et al., 1984*). We refer to the difference in G/C content at third positions and second positions of codons as 'G/C-skew', with positive G/C-skew expected for ORFs subject to purifying selection. We reasoned that we could exploit G/C-skew to assess the likelihood that novel ORFs identified by Ribo-RET have experienced purifying selection at the codon level. We assessed G/C skew for all 2299 novel ORFs identified in this study (leadered and leaderless). We limited the analysis to regions that do not overlap previously annotated genes, since G/C-skew could be impacted by selective pressure on an overlapping gene; 62% of ORFs were discarded because they completely overlap an annotated gene, and 17% of ORFs had some portion excluded. The set of all tested novel ORFs has modest, but significant, positive G/C-skew (Fisher's exact test $P < 2.2e^{-16}$;

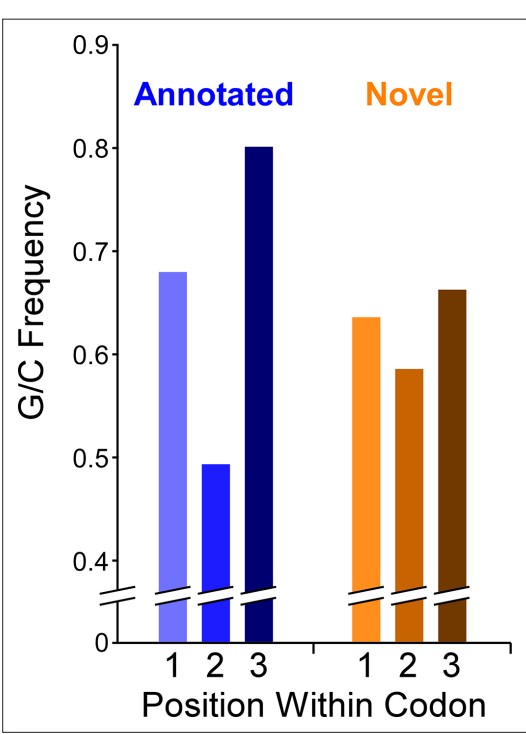

**Figure 7.** G/C skew within codons of novel and annotated ORFs. Histogram showing the frequency of G/C nucleotides at each of the three codon positions for annotated ORFs or novel ORFs. Note that only regions of novel ORFs that do not overlap a previously annotated ORF were analyzed.

n = 19,750 codons; *Figure 7*; *Supplementary file 1A, C* ), consistent with a subset of codons in this class having been under purifying selection. However, the degree of positive G/C-skew for the novel ORFs is much smaller than that for the annotated ORFs we identified in our datasets (*Figure 7*), suggesting that the proportion of novel ORFs experiencing purifying selection, and/or the intensity of that selection, is much lower than that for the annotated ORF group. To identify specific novel ORFs that have likely experienced purifying selection of their codons, and hence are likely to contribute to cell fitness, we determined G/C-skew for the non-overlapping regions of each novel ORF individually. We then ranked the ORFs by the significance of their G/C-skew (Fisher's exact test; see Materials and methods for details). Of the 103 ORFs with the most significant G/C-skew, there is a strong enrichment for positive G/C-skew: 90 of the ORFs have positive G/C skew and 13 have negative G/C skew. This suggests that ~80 of the 90 ORFs with positive G/C skew have been subject to purifying selection on their codons. It is important to note that the size of the ORF is a major consideration when determining the significance of G/C-skew; the small size of novel ORFs therefore limits this analysis. Moreover, the G/C-skew analysis provides no information on regions of novel ORFs that overlap annotated genes. Hence, the number of novel ORFs that we predict to be functional based on their G/C-skew is almost certainly a substantial underestimate. Nonetheless, the overall G/C-skew of novel ORFs relative to that of annotated ORFs provides strong evidence that the majority of novel ORFs are not functional.

## Discussion

### Ribo-Seq identifies thousands of isoform and novel ORFs

We have identified thousands of actively translated novel and isoform ORFs with high confidence. This conclusion is strongly supported by the clear association of initiating and terminating ribosomes with the start and stop codons, respectively, in untreated cells. We note that the enrichment of terminating ribosomes at the stop codons of novel ORFs in Ribo-seq data (i.e. no retapamulin treatment) is independent of the methods used to identify the novel ORF start codons. The novel and isoform ORFs are also supported by validation of selected ORFs using multiple independent genetic and biochemical approaches. Overall, our data reveal a far greater number of ORFs than previously appreciated, with annotated ORFs outnumbered by isoform and novel ORFs. Many genomic regions encode overlapping ORFs on opposite strands or on the same strand in different frames, contrary to the textbook view of genome organization.

There are 3898 annotated *Mtb* ORFs, but the ORF discovery approaches applied here undersampled these, identifying 1669. Failure to identify more annotated ORFs is likely due to the following biological and technical reasons: (i) Many genes are likely to be expressed at levels too low to be detected. In support of this idea, the median Ribo-seq read coverage for leadered, annotated ORFs identified by Ribo-RET was significantly higher than that for equivalent ORFs not identified by Ribo-RET (3.8-fold; Mann-Whitney U test $P < 2.2e^{-16}$); (ii) Many ORF start codons are likely to be misannotated, so they would be classified as isoforms. (iii) The A/T sequence preference of MNase (*Figure 2—figure supplement 2*) likely led to exclusion of some ORFs from the Ribo-RET libraries. In support of this idea, the base at position +17 relative to the start codon (i.e. immediately downstream of the preferred MNase cleavage site) is 1.7-fold more likely to be 'A', and 1.6-fold less likely to be 'G', for annotated ORFs we identified, than for those we did not. Given the clear underrepresentation of annotated ORFs in our datasets, we conclude that there are many more isoform and novel ORFs to be discovered.

### The abundance of novel start codons likely reflects pervasive translation

Evidence from other bacterial species suggests that the primary determinants of leadered translation initiation in *Mtb* are likely to be (i) a suitable start codon, (ii) an upstream sequence that can act as a S-D, and (iii) low local secondary structure around the ribosome-binding site. We detected enrichment of three different start codons in Ribo-RET data (*Figure 2B*), while S-D sequences can be located at a range of distances upstream of the start codon (*Vellanoweth and Rabinowitz, 1992*). Hence, there is limited sequence specificity associated with translation initiation. Moreover, a recent report showed that in *E. coli*, an S-D sequence is not an essential requirement for translation initiation (*Saito et al.,*

*2020*). Leaderless translation initiation has even fewer sequence requirements; our data suggest that a 5' AUG or GUG is sufficient for robust leaderless translation (*Shell et al., 2015*). While AUG and GUG represent only ~3% of all possible trinucleotide sequences, there is likely to be a strong bias towards 5' AUG or GUG from the process of transcription initiation; the majority of TSSs in *Mtb* are purines, and the majority of +2 nucleotides are pyrimidines (*Cortes et al., 2013*; *Shell et al., 2015*). We propose that many *Mtb* transcripts are subject to spurious translation either by the leaderless or leadered mechanism, simply because the nominal sequence requirements for these processes commonly occur by chance. Thus, there is pervasive translation of the *Mtb* transcriptome, similar to the pervasive translation described in eukaryotes (*Ingolia et al., 2014*; *Ruiz-Orera et al., 2018*; *Wacholder et al., 2021*). Pervasive translation has been proposed as an explanation for some of the novel ORFs detected in *E. coli* by Ribo-RET (*Meydan et al., 2019*).

The process of pervasive translation is analogous to pervasive transcription, whereby many DNA sequences function as promoters, often from within genes, to drive transcription of spurious RNAs (*Lybecker et al., 2014*; *Wade and Grainger, 2014*). Indeed, there are many intragenic promoters in *Mtb* (*Cortes et al., 2013*; *Shell et al., 2015*), providing an additional source of potential spurious translation. We speculate that like spurious transcripts, which are rapidly degraded by RNases, the protein products of pervasive translation are rapidly degraded, as has been proposed for pervasively translated ORFs in *E. coli* (*Stringer et al., 2021*). Since Ribo-seq and Ribo-RET detect translation, not the protein product, the stability of the encoded proteins would not impact our ability to detect the corresponding ORFs.

Pervasive translation, by definition, means that ribosomes will spend some fraction of the time translating spurious ORFs. Although we detected many more novel ORFs than annotated ORFs, the total number of codons in all detected novel ORFs is ~20% that of annotated ORFs because of the smaller size of novel proteins. Moreover, novel ORFs as a group are expressed at substantially lower levels than annotated ORFs (*Figure 5*; *Figure 5—figure supplements 1 and 2*). Thus, it is likely that <10% of translation in *Mtb* at any given time is of spurious ORFs, so pervasive translation is unlikely to be overly detrimental to the cell.

## Proto-genes and the evolution of new functional genes

Studies of eukaryotes indicate the existence of proto-genes, targets of pervasive translation of either intergenic sequences or sequences antisense to annotated genes (*Ingolia et al., 2014*; *Ruiz-Orera et al., 2018*; *Wacholder et al., 2021*). Proto-genes have the potential to evolve into functional ORFs that contribute to cell fitness (*Blevins et al., 2021*; *Carvunis et al., 2012*; *Lu et al., 2017*; *Ruiz-Orera et al., 2018*; *Vakirlis et al., 2018*; *Vakirlis et al., 2020*; *Van Oss and Carvunis, 2019*). There is also evidence that some bacterial protein-coding genes evolved from intergenic sequence (*Yomtovian et al., 2010*). Our data suggest that *Mtb* has a rich source of proto-genes. As described for proto-genes in yeast, the novel ORFs we identified in *Mtb* tend to be less well expressed, have less adapted codon usage, and are shorter than annotated genes (*Blevins et al., 2021*; *Carvunis et al., 2012*). Pervasive translation in *Mtb* likely facilitates the evolution of new gene function in *Mtb*. Since pervasive translation represents a low proportion of all translation, the fitness cost of pervasive translation may be balanced by the benefits of having a large pool of proto-genes.

## New functional ORFs/proteins in *Mtb*

The question of whether an ORF is functional first requires a definition of function (*Keeling et al., 2019*). Here, we define function as the ability to improve cell fitness. While functional ORFs need not be under purifying selection, ORFs undergoing purifying selection are presumably functional. One metric of purifying selection available in the G/C-rich genomes of mycobacteria is G/C-skew. Analysis of G/C-skew in the codons of novel ORFs identified 90 ORFs that are likely to be functional (positive G/C, p < 0.1 in *Supplementary file 1A, C*). 54 of these 90 novel ORFs are leadered, and the Ribo-RET signal associated with these 54 ORFs was significantly higher than that for the set of all other novel ORFs (Mann-Whitney U test $P = 1.8e^{-5}$), consistent with the idea that functional ORFs are likely to be more highly expressed than non-functional ORFs (*Carvunis et al., 2012*; *Vakirlis et al., 2020*). Of the 90 ORFs that are likely functional based on their G/C-skew, 44 are ≤51 codons long. Thus, this single indicator of purifying selection has greatly expanded the set of likely functional small ORFs/proteins described for *Mtb*. There may be other constraints that additionally limit codon selection, especially

for regulatory sORFs, such that functional sORFs lack positive G/C skew. Indeed, this is the case for a phylogenetically conserved set of cysteine-rich regulatory sORFs; cysteine codons that are likely to be essential for sORF regulatory function (*Canestrari et al., 2020*) also reduce the G/C-skew (*Supplementary file 1D*).

Analysis of codon usage for isoform ORFs is not informative due to their overlap with annotated ORFs. Some isoform ORFs are likely to represent mis-annotations of annotated ORFs. Multiple lines of evidence support this idea: (i) 19% (288) of isoform start codons are ≤10 codons from the corresponding annotated start codon (*Supplementary file 1E*); this was 3.4-fold more likely for leaderless isoform ORFs, presumably because they lack a S-D, which likely reduces the accuracy of start codon prediction by annotation pipelines. (ii) Leadered isoform ORFs that initiate within 10 codons of an annotated ORF have significantly higher Ribo-RET occupancy than other leadered isoform ORFs (Mann Whitney U Test $P = 6.3e^{-13}$; Ribo-RET occupancy from a single replicate), and are significantly less likely to overlap an annotated gene whose start codon was identified by Ribo-RET (Fisher's Exact Test $P = 3e^{-4}$). Nonetheless, since most isoform ORFs start far from an annotated ORF start, we presume that most do not represent mis-annotations; indeed, for 43% (644) of the isoform ORFs, we also detected the start codon of the overlapping annotated ORF by Ribo-RET. While we expect many isoform ORFs to be a manifestation of pervasive translation, we speculate that some encode proteins with functions related to the function of protein encoded by the overlapping, annotated gene, as has been proposed for isoform ORFs in *E. coli* (*Meydan et al., 2019*).

## Conclusions

Our data suggest that the *Mtb* transcriptome is pervasively translated. The unprecedented extent of translation we observe suggests that much of the translation is biological 'noise', and that most of the translated ORFs are unlikely to be functional. As ribosome-profiling studies are extended to more diverse species, we anticipate a massive increase in the discovery of bacterial sORFs/small proteins. Future studies aimed at functional characterization of sORFs/small proteins will require prioritizing with clear supporting evidence for function from codon usage patterns, phylogenetic conservation (*Sberro et al., 2019*), or genetic data.

## Materials and methods

### Key resources table

| Reagent type (species) or resource | Designation | Source or reference | Identifiers | Additional information |
|---|---|---|---|---|
| Strain, strain background (*Mycobacterium tuberculosis*) | mc²7000 | DOI: 10.1016/j.vaccine.2006.05.097 | Δ*panCD* Δ*RD1* | |
| Strain, strain background (*Mycobacterium smegmatis*) | mc²155 | DOI: 10.1111/j.1365–2958.1990.tb02040.x | | |
| Antibody | Monoclonal anti-FLAG M2 antibody (Mouse monoclonal) | SIGMA | Catalog # F1804 | Used at (1:1,000) dilution for western blot |
| Recombinant DNA reagent | pRV1133C (plasmid) | This study | pRV1133C | Integrates at *attP* site; includes the *metE* promoter region |
| Recombinant DNA reagent | pGE450 (plasmid) | This study | pGE450 | Derivative of pRV1133C containing 3 x FLAG |
| Recombinant DNA reagent | pGE190 (plasmid) | This study | pGE190 | Derivative of pRV1133C containing the nLuc gene from pNL1.1 (Promega, cat no 1001) |
| Other | Micrococcal nuclease (S7) | SIGMA | Catalog # 10107921001 | |
| Other | Nano-Glo Luciferase Assay Reagent | Promega | Catalog # N1110 | |
| Chemical compound, drug | Retapamulin | SIGMA | Catalog # CDS023386 | |

*Continued on next page*

*Continued*

| Reagent type (species) or resource | Designation | Source or reference | Identifiers | Additional information |
|---|---|---|---|---|
| Software, algorithm | CLC Genomics Workbench | Qiagen | v8.5.1 | Alignment of sequence reads from. fastq files |
| Software, algorithm | RNAfold | DOI:10.1186/1748-7188-6-26 | v2.4.14 | ViennaRNA Package https://www.tbi.univie.ac.at/RNA/ |

## Strains and plasmids

All oligonucleotides used in this study are listed in *Supplementary file 1F*. Ribo-seq and Ribo-RET experiments were performed using the *M. tuberculosis* strain mc²7000 (*Sambandamurthy et al., 2006*). *M. tuberculosis* mc²7000 cells were grown in 7H9 medium supplemented with 10% OADC (Oleic acid, Albumin, Dextrose, Catalase), 0.2% glycerol, 100 µg/ml pantothenic acid and 0.05% Tween80 at 37 °C, without shaking, to an $OD_{600}$ of ~1.

We constructed a shuttle vector, pRV1133C, to allow integration of luciferase or FLAG-tag fusion constructs into the *M. smegmatis* mc²155 (*Snapper et al., 1990*) chromosome, with a constitutive promoter driving transcription. pRV1133C was derived from pMP399, retaining its *oriE* for episomal maintenance in *E. coli*, its integrase and *attP* site for integration at the L5 *attB* site in mycobacteria, and apramycin resistance (*Consaul and Pavelka, 2004*). The *hsp60* promoter of pMP399 was replaced by the promoter of the *M. tuberculosis Rv1133c* (*metE*) gene (genome coordinates 1,261,811–1,261,712 from the minus strand of the *M. tuberculosis* genome, stopping one base pair upstream of the transcription start site; GenBank accession: AL123456.3). The criterion for selecting *Rv1133c* was its strong constitutive expression assessed by transcription start site metrics (*Shell et al., 2015*).

A luciferase (NanoLuc) gene amplified from pNL1.1 (Promega, cat. no 1001) was cloned downstream of the *Rv1133c* promoter to generate pGE190. To construct individual reporter fusion plasmids, the entire pGE190 plasmid was amplified by inverse PCR using Q5 High Fidelity DNA polymerase (NEB) with oligonucleotides TGD4006 and TGD5162. Sequences corresponding to the 25 bp upstream of, and including the start codons for selected ORFs were PCR-amplified using oligonucleotide pair TGD5163 and TGD5164, to amplify template oligonucleotides TGD5165-5173, TGD5175, TGD5178-5186, and TGD5795-5797. PCR products were cloned into the linearized pGE190 using the In-Fusion cloning system (Takara). The oligonucleotide templates had a 'Y' (mixed base 'C' or 'T') at the position corresponding to the central position of the start codon. Clones were sequenced to identify wild-type and mutant constructs, where the central position of the start codon was a 'T' or a 'C', respectively. Plasmid DNA was electroporated into *M. smegmatis* mc²155 for chromosomal integration before assaying luciferase activity.

A 3 x FLAG-epitope-tag sequence was integrated into pRV1133C to generate pGE450. To construct individual FLAG-tagged constructs, the entire pGE450 plasmid was amplified by inverse PCR using Q5 High Fidelity DNA polymerase (NEB) with oligonucleotides TGD4981 and TGD4982. Sequences from the predicted transcription start site up to the stop codon for selected ORFs were PCR-amplified using oligonucleotide pairs TGD5208 and TGD5209, TGD5216 and TGD5217, TGD5241 and TGD5242, or TGD5247 and TGD5248. PCR products were cloned into pGE450 using the In-Fusion cloning system (Takara). Start codon mutant constructs were made by inverse PCR-amplification of the wild-type constructs using primers that introduce a start codon mutation ('T' to 'C' change at the central position of the start codon; oligonucleotides TGD5210, TGD5211, TGD5218, TGD5219, TGD5256, TGD5257, TGD5258, and TGD5259). PCR products were treated with *Dpn*I and cloned using the In-Fusion cloning system (Takara). Following sequence confirmation, plasmid DNA was electroporated into *M. smegmatis* mc²155 for chromosomal integration before performing expression analysis by western blot.

## Ribo-seq without drug treatment

Ten ml of *M. tuberculosis* ($OD_{600}$ of 0.4) was used to inoculate 400 ml of medium and grown to an $OD_{600}$ of 1 (2–3 weeks). Cells were collection by filtration through a 0.22 µm filter. Libraries were prepared for sequencing, and sequencing data were processed as described previously for *M. smegmatis* (*Shell et al., 2015*).

## RNA-Seq

Cell extracts were prepared in parallel to those used for Ribo-seq. RNA was extracted using acid phenol and chloroform followed by isopropanol precipitation. Ribosomal RNA was removed using the Ribo-Zero Magnetic Kit (Epicentre). RNA fragmentation, library preparation, sequencing, and data processing were performed as described previously for *M. smegmatis* (*Shell et al., 2015*).

## Ribosome profiling with retapamulin treatment (Ribo-RET)

Ten ml of *M. tuberculosis* (OD$_{600}$ of 0.4) was used to inoculate 400 ml of medium and grown to an OD$_{600}$ of 1 (2–3 weeks). Cells were treated with retapamulin (Sigma CDS023386) at a final concentration of 0.125 mg/ml for 15 min at room temperature, with occasional manual shaking, and collected by filtration through a 0.22 µm filter. Cells were flash frozen in liquid nitrogen with 0.7 ml lysis buffer (20 mM Tris pH 8.0, 10 mM MgCl$_2$, 100 mM NH$_4$Cl, 5 mM CaCl$_2$, 0.4% Triton X-100, 0.1 % NP-40, 1 mM chloramphenicol, 100 U/mL DNase I). Frozen cells were milled using a Retsch MM400 mixer mill for 8 cycles of 3 min each at 15 Hz. Milling cups were re-frozen in liquid nitrogen in between each milling cycle. Cell extracts were thawed and incubated on ice for 30 min. Samples were clarified by centrifugation. Supernatants were passed twice through 0.22 µm filters. 1 mg aliquots of cell extracts were flash-frozen in liquid nitrogen. Monosomes were isolated by digesting 1 mg of cell extract with 1,500 units of micrococcal nuclease for 1 hr at room temperature on a rotisserie rotator. The reaction was quenched by adding 2 µl 0.5 M EGTA, after which the digest was fractionated through a 10–50% sucrose gradient. Fractions from the sucrose gradients were electrophoresed on a 1% agarose gel with 1% bleach to identify ribosomal RNA peaks. Those fractions were selected, pooled, and monosomes isolated by acid phenol:chloroform extraction and isopropanol precipitation.

Libraries for sequencing were prepared using a previously described method (*Ingolia, 2010*). RNA from monosomes was run on a 15% denaturing gel alongside a 31 nt RNA oligonucleotide to size-select 31 ± 5 nt fragments. Samples were gel-extracted in 500 µl RNA gel extraction buffer (300 mM NaOAc (pH 5.5), 1 mM EDTA, 0.1 U/mL SUPERase-In RNase inhibitor) followed by isopropanol precipitation. The samples were dephosphorylated by incubating with 10 U of T4 Polynucleotide Kinase (NEB) for 1 hr at 37 °C, before extraction with phenol:chloroform:isoamyl alcohol, and ethanol precipitation. The dephosphorylated RNAs were ligated to the 3' linker oligonucleotide JW9371 using T4 RNA Ligase 2 (truncated, K227Q) at a 1:4 RNA:linker ratio. The ligation reactions were incubated for 3 hr at 37 °C, followed by 20 min at 65 °C. The reactions were separated on a 15% polyacrylamide denaturing gel alongside a control RNA oligonucleotide (JW9370) of the expected size of the ligated product. The RNA-ligation products were excised and extracted in 500 µL RNA extraction buffer and concentrated by ethanol precipitation. Reverse transcription was performed on the RNA samples using Superscript III (Life Technologies) and oligo JW8875, as described previously (*Ingolia, 2010*). The reactions were separated through a 10% polyacrylamide denaturing gel and cDNAs excised and extracted in 500 µL DNA extraction buffer (300 mM NaCl, 10 mM Tris-Cl (pH 8), 1 mM EDTA). Reverse-transcribed cDNA was circularized using CircLigase, and PCR-amplified as described previously (*Ingolia, 2010*). Between 4 and 9 cycles of PCR were performed using Phusion High Fidelity DNA Polymerase, JW8835 as the standard forward primer, and JW3249, 3250, 8876 or 8877, corresponding to Illumina index numbers 1, 2, 34, or 39 respectively, as the reverse primer. Samples were separated through an 8% polyacrylamide gel. DNAs of the appropriate length (longer than the control adapter band) were excised from the gel and extracted in 500 µL of DNA extraction buffer. DNAs were concentrated by isopropanol precipitation. Samples were quantified and subject to DNA sequence analysis on a NextSeq instrument.

## Inferring ORF positions from Ribo-RET data

Sequencing reads from Ribo-RET datasets were trimmed to remove adapter sequences using a custom python script that trimmed reads up to the first instance of CTGTAGGCACC, keeping trimmed reads in the length range 20–44 nt. Trimmed sequence reads were aligned to the reference genome and separately to a reverse-complemented copy of the reference genome, using Rockhopper (*McClure et al., 2013*). The positions of read 3' ends were determined from the resultant.sam files, and used to determine coverage on each strand at each genome position. Coverage values were set to 0 for regions encompassing all annotated non-coding genes, and the 50 nt regions downstream of the 1285 TSSs associated with an RUG (i.e. the first 50 nt of all predicted leaderless ORFs) (*Cortes et al.,*

*2013*; *Shell et al., 2015*). Read counts were then normalized to total read count as reads per million (RPM).

Every genome coordinate on each strand was considered as a possible IERF. To be selected as an IERF, a position required a minimum of 5.5 RPM coverage (equivalent to 20 sequence reads in the first replicate dataset), with at least 10-fold higher coverage than the average coverage in the 101 nt region centered on the coordinate being considered, and equal or higher coverage than every position in the 21 nt region centered on the coordinate being considered. For high-confidence ORF calls, all criteria had to be met in both replicate datasets. IERFs were inferred to represent an ORF if the IERF position was 15 nt downstream of a TTG, 14–18 nt downstream of an ATG, or 14–18 nt downstream of a GTG. These trinucleotide sequences and distances were selected based on a > 1.4 fold enrichment upstream of IERFs (*Figure 2B*). In a small number of cases, two IERFs were associated with the same start codon; this only occurred in cases where the two IERFs had identical Ribo-RET sequence coverage. This double-matching means that the number of IERFs is slightly higher than the number of identified ORFs.

## Calculating false discovery rates for ORF prediction from Ribo-RET data

The likelihood of randomly selecting a genome coordinate with an associated start codon sequence (as defined above for IERFs) was estimated by selecting 100,000 random genome coordinates and determining the fraction, 'R', that would be associated with a start codon. The set of IERFs contains a number of true positives (i.e. corresponding to a genuine start codon), and a number of false positives. We assume that true positive IERFs are all associated with a start codon using the parameters described above for calling ORFs. We assume that false positive IERFs are associated with a start codon at the same frequency as random genome coordinates, that is R. Since we know how many IERFs were not associated with a start codon, we can use this number to estimate how many false positive IERFs were associated with a start codon by chance. With the total number of IERFs as 'I' and the total number of identified ORFs as 'O', the FDR for ORF calls is estimated by:

$$(100^*(I - O)^*(R/(1 - R)))/O$$

To estimate the distribution of false positive IERFs between annotated, isoform, and novel ORFs, we determined the relative proportion of each class of ORF from the set of randomly selected genome coordinates that were associated with a start codon by chance.

## Selection of mock ORFs

As a control for potential artifacts of DNA sequence on Ribo-seq coverage we selected 1,000 mock ORFs: sequences that begin at an ATG or GTG and extend to the first in-frame stop codon. Mock ORF stop codons do not match those of previously annotated genes or novel genes identified from Ribo-RET data. To ensure that mock ORFs are in transcribed regions, we required non-zero RNA-seq coverage at the first position of each mock ORF. For simplicity, mock ORFs were only selected on the forward strand of the genome.

## RNA folding prediction

The sequence from –40 to +20 relative to each start codon, or for 500 × 60 nt sequences randomly selected from the *M. tuberculosis* genome, were selected for prediction of the free energy of the predicted minimum free energy structure using a local installation of ViennaRNA Package tool RNAfold, version 2.4.14, using default settings (*Lorenz et al., 2011*).

## Determining normalized sequence read coverage from Ribo-Seq and RNA-Seq data

Library construction for Ribo-seq and RNA-seq included polyadenylation of RNA fragments, and sequence reads were trimmed at their 3′ ends, immediately upstream of the first instance of 'AAA', before aligning to the reference genome; hence, it is impossible for a trimmed sequence read to end with an 'A'. This likely explains why we observed apparent differences in ribosome occupancy in Ribo-seq data precisely at start and stop codons for all classes of ORF (e.g. *Figure 1*), since these codons are strongly enriched for specific bases. We note that the same patterns were observed for

RNA-seq data (*Figure 4—figure supplement 1B-D*; RNA-seq library construction included a polyadenylation step, and reads were trimmed and mapped identically to those from Ribo-seq datasets), and for mock ORFs in Ribo-seq data (*Figure 4—figure supplement 1A*).

Sequence reads were aligned to the reference genome (NCBI Reference Sequence: NC_000962.3) and separately to a reverse-complemented copy of the reference genome, using Rockhopper, version 2.0.3 (*McClure et al., 2013*). The positions of read 3′ ends were determined from the resultant .sam files, and used to determine coverage on each strand at each genome position, normalized to total read count as reads per million (RPM).

## Generating metagene plots

Metagene plots (i.e. *Figure 1A–D*, *Figure 1—figure supplement 1*, *Figure 2A*, *Figure 2—figure supplement 1*, *Figure 4B–D*, *Figure 4—figure supplement 1A-D*; *Figure 4—figure supplement 2D-F*) used normalized coverage values (RPM) for Ribo-seq, RNA-seq, or Ribo-RET data, calculated as described above. Coverage scores were selected for regions from –50 to +100 relative to start codons or TSSs, or –100 to +50 relative to stop codons. Coverage RPM values were further normalized to the highest value in the selected range. For metagene plots of leadered, previously annotated ORFs (*Figures 1A and 2A*, *Figure 2—figure supplement 1*), previously annotated genes were excluded if they were pseudogenes, non-coding, or had a TSS within 5 nt upstream of the start codon. For the metagene plot of TSSs not associated with leaderless ORFs, TSSs were selected from published reports (*Cortes et al., 2013*; *Shell et al., 2015*) if they were located at least 6 nt upstream of a previously annotated start codon.

## Calculating relative ribosome density per transcript for ORFs and transcript regions

We selected three sets of genomic regions: (i) all annotated ORFs identified either by Ribo-RET (higher confidence and lower confidence) or from leaderless analysis, (ii) all novel ORFs identified either by Ribo-RET (higher confidence and lower confidence) or from leaderless analysis, and (iii) a set of 1854 control transcript regions, described below. For (ii), we removed regions of ORFs that are not at least 30 nt from an annotated gene on the same strand; in many cases this led to exclusion of the ORF or trimming one or both ends of the region to be analyzed. We also excluded any remaining ORF or ORF region <50 nt in length. A set of control transcript regions, intended to comprise mostly non-coding RNA, was selected by identifying transcription start sites (*Cortes et al., 2013*; *Shell et al., 2015*) > 5 nt upstream of an RTG trinucleotide sequence. We then selected the first 50 nt of the associated transcribed regions. These control regions were excluded if they are not at least 30 nt from an annotated gene on the same strand, or if they overlap partially or completely a novel or isoform ORF identified in this study.

For each category of region, (i), (ii), and (iii), described above, we calculated the normalized sequence read coverage (RPM) from two replicates each of RNA-seq and Ribo-seq data, aligning only the sequence read 3′ ends (see section titled 'Determining normalized sequence read coverage from Ribo-seq and RNA-seq data'). We excluded 7 of the regions in category (iii) that had zero RNA-seq coverage in both replicates. Data in *Figure 5A* and *Figure 5—figure supplement 1*, show the sequence read coverage normalized to the length of each region analyzed. To calculate the relative ribosome occupancy per transcript (*Figure 5B*), we first averaged the RNA-seq and Ribo-seq normalized coverage values from each of the two replicate datasets for each region analyzed. We then calculated the ratio of the Ribo-seq value to the RNA-seq value.

## Analysis of G/C usage within codons

For novel ORFs identified using the first replicate of Ribo-RET data, including ORFs identified in both replicates, we first trimmed the start and stop codons. We then trimmed any region of the remaining ORF that overlaps a previously annotated gene, leaving only complete codons; in many cases this removed the entire ORF from the analysis. For the remaining sequences, we scored the first, second or third position of all codons for the presence of a G or C. The G/C-skew was calculated as the ratio of the sum of G/C bases at the third codon position to that at the second codon position. Statistical comparisons were performed using a Fisher's exact test comparing G/C base count at the second and third positions; tests were one-tailed or two-tailed as indicated, with the null hypothesis for one-tailed

tests being that the G/C base count at the third codon position was not higher than that at the second codon position. Values plotted in *Figure 7* represent the sum of values for each individual ORF, or the equivalent number for the annotated ORFs we identified by Ribo-RET (we did not trim these except for start and stop codons).

## Analysis of G/C usage within codons for predicted regulatory cysteine-rich sORFs

We examined the G/C-skew of 6 ORFs that we predict regulate expression of the downstream gene in response to cysteine availability, based on their conservation with regulatory sORFs in *M. smegmatis* (*Canestrari et al., 2020*). Strikingly, only one of these ORFs individually has significantly positive G/C-skew (*Supplementary file 1D*; Fisher's exact test $p < 0.05$). Moreover, as a group, the six sORFs do not have significantly positive G/C-skew (*Supplementary file 1D*; Fisher's exact test $p = 0.13$; $n = 145$ codons). We repeated this analysis after removing the cysteine codons from the sORFs, reasoning that cysteine codons have a neutral or negative effect on G/C-skew, and that the presence of cysteine codons is likely essential for the regulatory activity of the sORFs. Removing the cysteine codons increased G/C-skew for all ORFs; in two cases, the G/C-skew of the ORFs with cysteine codons removed is significantly positive (*Supplementary file 1D*; Fisher's exact test $p < 0.05$). Moreover, as a group, the six ORFs with cysteine codons removed have significantly positive G/C-skew (*Supplementary file 1D*; Fisher's exact test $p = 1.5e^{-4}$; $n = 120$ codons).

## Analysis of trinucleotide sequence content upstream of IERFs

The frequency of each trinucleotide was determined for the 50 nt upstream of all IERFs. For each trinucleotide sequence, the frequencies at positions –50 to –41 were averaged (mean), and frequencies at all other positions were normalized to this averaged number. The frequency of AGG and GGA trinucleotide sequences upstream of putative start codons was determined similarly, with the control region used for normalization located at positions –35 to –26 relative to the start codons.

## Luciferase reporter assays

*M. smegmatis* mc²155 strains with integrated luciferase reporter constructs were grown in TSB with Tween80 overnight at 37 °C to an $OD_{600}$ of ~1.0. Ten µl of Nano-Glo Luciferase Assay Reagent was mixed with 10 µl of cell culture. Luminescence readings (relative light units; RLUs) were taken using a Turner Biosystems Veritas microplate luminometer. Relative luminescence values were reported as $RLU/OD_{600}$. Assays were performed in triplicate (biological replicates).

## Western blots

*M. smegmatis* MC²155 strains with integrated FLAG-tagged constructs were grown in TSB with Tween overnight at 37 °C to an $OD_{600}$ of ~1.0. Cells were harvested by centrifugation and resuspended in 1 x NuPage LDS sample buffer (Invitrogen) +5 mM sodium metabisulfite. Samples were heated at 95 °C for 10 min before loading onto a 4–12% gradient Bis-Tris mini-gel (Invitrogen). After separation, proteins were transferred to a nitrocellulose membrane (Life Technologies) or a PVDF membrane (Thermo Scientific). Membranes were probed with a monoclonal mouse anti-FLAG antibody (M2; Sigma). Secondary antibody and detection reagents were obtained from Lumigen (ECL plus kit) and used according to the manufacturer's instructions.

## Integrated genome browser

All ribosome profiling and Ribo-RET data, and identified ORFs are available for visualization on our interactive genome browser (*Shell et al., 2015*): https://mtb.wadsworth.org/.

## Mass spectrometry

Five ml of *Mtb* ($OD_{600}$ of 0.4) was used to inoculate 200 ml of medium and grown to an $OD_{600}$ of 1.175. Cells were collection by filtration through a 0.22 µm filter. Cells were flash frozen in liquid nitrogen with 0.6 ml lysis buffer (20 mM Tris pH 8.0, 10 mM $MgCl_2$, 100 mM $NH_4Cl$, 5 mM $CaCl_2$, 0.4% Triton X100, 0.1 % NP-40, 1 mM chloramphenicol, 100 U/mL DNase I). Frozen cells were milled using a Retsch MM400 mixer mill for 8 cycles of 3 min each at 15 Hz. Milling cups were re-frozen in liquid nitrogen between each milling cycle. Cell extracts were thawed and incubated on ice for 30 min. Samples were

clarified by centrifugation. Supernatants were passed twice through 0.22 µm filters. Samples were prepared for MS analysis from 100 µg aliquots of *Mtb* cytosolic lysate in each of six different ways, with the numbers listed below matching those in *Supplementary file 1A, C*:

1. Protein was precipitated by addition of acetonitrile at a ratio of 2:1, placed on ice for 20 min, then clarified by centrifugation for 10 min at 12,000 x g. The supernatant (enriched for small proteins) was decanted into two aliquots and dried using a speedvac (Thermo). One aliquot was resuspended and using a 10 mg HLB Solid-phase extraction cartridge (Waters) according to the manufacturer's instructions, and dried. The second aliquot was used in method (2), as described below.

2. The remaining aliquot from (1) was resuspended in 100 µl Tri-ethyl ammonium bicarbonate TEAB (Sigma) and subjected to dimethyl labeling of Lys and N-termini to increase the mass and reduce the charge, and thereby increase detectability of small proteins (*Boersema et al., 2009*; *Yan et al., 2020*). The sample was desalted as above, and dried.

3. Protein was denatured by addition of powdered urea (Alfa Aesar) to 8 M final concentration. The sample was subjected to centrifugal filtration through a 10 K Amicon filter similar to that employed in Filter Aided Sample Prep (FASP) proteomics (*Wisniewski et al., 2009*), except the sample flow-through containing small molecular weight proteins, not the retentate, was retained and split into two aliquots. One aliquot was desalted and dried. The second aliquot was used in method (4), as described below.

4. The remaining aliquot from (3) was diluted until the urea concentration was <2 M, before being chemically reduced and alkylated (*Yan et al., 2020*). To reduce the size of large and hydrophobic proteins, the sample was digested with 1 µg of sequencing-grade trypsin (Promega) for 6 hr at 37 °C. Following digestion, the sample was quenched by addition of formic acid, desalted and dried. Samples were resuspended in 20 µl of 0.2% formic acid in water.

5. Protein was denatured by addition of powdered urea (Alfa Aesar) to 8 M final concentration. The sample was subjected to centrifugal filtration through a 3 K Amicon filter, retaining the small molecular weight protein flow-through. The sample was desalted and dried.

6. A total-protein digest was performed using an in-solution trypsin digestion procedure, as a potential source for small proteins not enriched using the approaches described above (*Champion et al., 2003*).

All samples were analyzed by nano-UHPLC-MS/MS on a Q-Exactive instrument (Thermo) (*Bosserman et al., 2017*; *Canestrari et al., 2020*). RAW files were converted to mgf (mascot generic format) using MS-Convert (*Adusumilli and Mallick, 2017*). Spectrum mass matching was performed using the Paragon Algorithm with feature sets as appropriate for each sample (e.g. demethylation, trypsin, no-digest) in thorough mode (*Champion et al., 2012*; *Shilov et al., 2007*). A custom small protein, and leaderless FASTA constructed from the Ribo-seq data was used for database search. FDRs were determined using the target-decoy strategy, as in *Elias and Gygi, 2007*. Proteins identified using this method were subjected to manual spectral interpretation to validate peptide spectral matches, in particular for b,y-ion consistency, and y fragments to Pro with high intensity. The presence of His and Phe immonium ions (110.0718, 120.0813 [M + H] + m/z), when present in the target sequence, were used for additional validation. Selected small proteins and small protein-derived peptides from high- and medium-observed abundance proteins were chemically synthesized (Genscript) and subjected to LC-MS/MS as above. Synthetic small protein spectra were compared to the empirical-matched small proteins using the peptide spectral annotator (*Brademan et al., 2019*).

## Acknowledgements

We thank Mike Palumbo and Dan Muller for assistance setting up the interactive genome browser (https://mtb.wadsworth.org/), Gabriele Baniulyte, Yunlong Li, and Yong Yang for technical support, David Grainger for comments on the manuscript, and Anne-Ruxandra Carvunis for helpful discussions. We thank the Wadsworth Center Applied Genomic Technologies, Bioinformatics, and Media Core Facilities, and Dr. Boggess in the Notre Dame MS and Proteomics Facility. This work was supported by National Institutes of Health grants R21AI117158 and R21AI119427 (JTW, KMD, TAG) and R01GM139277 (JTW, KMD, TAG, MMC).

## Additional information

### Competing interests

Joseph T Wade: Reviewing editor, eLife. The other authors declare that no competing interests exist.

### Funding

| Funder | Grant reference number | Author |
|---|---|---|
| National Institute of Allergy and Infectious Diseases | R21AI117158 | Keith M Derbyshire<br>Todd A Gray<br>Joseph T Wade |
| National Institute of Allergy and Infectious Diseases | R21AI119427 | Keith M Derbyshire<br>Todd A Gray<br>Joseph T Wade |
| National Institute of General Medical Sciences | R01GM139277 | Matthew M Champion<br>Keith M Derbyshire<br>Todd A Gray<br>Joseph T Wade |

The funders had no role in study design, data collection and interpretation, or the decision to submit the work for publication.

### Author contributions

Carol Smith, Investigation, Writing – review and editing; Jill G Canestrari, Investigation; Archer J Wang, Investigation, Methodology; Matthew M Champion, Funding acquisition, Investigation, Project administration, Writing – review and editing; Keith M Derbyshire, Todd A Gray, Conceptualization, Funding acquisition, Project administration, Supervision, Writing – review and editing; Joseph T Wade, Conceptualization, Formal analysis, Funding acquisition, Methodology, Project administration, Software, Supervision, Visualization, Writing – original draft, Writing – review and editing

### Author ORCIDs

Keith M Derbyshire http://orcid.org/0000-0003-3404-8312
Joseph T Wade http://orcid.org/0000-0002-9779-3160

### Decision letter and Author response

Decision letter https://doi.org/10.7554/eLife.73980.sa1
Author response https://doi.org/10.7554/eLife.73980.sa2

## Additional files

### Supplementary files

• Supplementary file 1. Supplementary tables. (A) List of putative leaderless ORFs. (B) List of IERFs. (C) List of ORFs identified by Ribo-RET. (D) Analysis of G/C skew for cys-rich regulatory ORFs. (E) Analysis of isoform ORFs and their position relative to overlapping annotated ORFs. (F) List of oligonucleotides used in this study.

• Transparent reporting form

### Data availability

Raw Illumina sequencing data are available from the ArrayExpress and European Nucleotide Archive repositories with accession numbers E-MTAB-8039 and E-MTAB-10695. Raw mass spectrometry data are available through MassIVE, with exchange #MSV000087541. Python code is available at https://github.com/wade-lab/Mtb_Ribo-RET (copy archived at swh:1:rev:c6a41047e001550aab663588a13fe935547b9431).

The following datasets were generated:

| Author(s) | Year | Dataset title | Dataset URL | Database and Identifier |
|---|---|---|---|---|
| Smith C, Wang AJ, Wade J | 2019 | Pervasive Translation in Mycobacterium tuberculosis | https://www.ebi.ac.uk/arrayexpress/experiments/E-MTAB-8039/ | ArrayExpress, E-MTAB-8039 |
| Wang AJ, Wade J | 2021 | Pervasive Translation in Mycobacterium tuberculosis | https://www.ebi.ac.uk/arrayexpress/experiments/E-MTAB-10695/ | ArrayExpress, E-MTAB-10695 |

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
