## [Editor Report]

The use of ribosome profiling in this study allowed for the identification of translated regions of the *Mycobacterium tuberculosis* genome, identifying new genomic regions that undergo active translation. A select set of these appears to have been the subject of purifying evolutionary selection, suggesting that this pervasive translation of short genetic regions serves as the basis for the evolution of new proteins/protein functions.

---

## [Decision Letter]

**Decision letter after peer review:**

[Editors’ note: the authors submitted for reconsideration following the decision after peer review. What follows is the decision letter after the first round of review.]

Thank you for submitting your work entitled "Pervasive translation in *Mycobacterium tuberculosis*" for consideration by *eLife*. Your article has been reviewed by 3 peer reviewers, and the evaluation has been overseen by me serving as both Reviewing and Senior Editor. The reviewers have opted to remain anonymous.

As you can see from their comments below, the reviewers were somewhat discrepant in their perspectives on this paper, necessitating more extensive consultation. The major concern of two of the reviewers was that ~1000 novel ORFs seems extremely high. Both of these reviewers thought further computational analysis (such as re-evaluation of the ribosome profiling data to better sort out real candidates from noise) and further validation (such as tagging of representative candidates in the native genome) was required to support this conclusion. Given the centrality of the "pervasive translation" to your paper, I agree that further analysis and validation is warranted. Since it is anticipated that this work will take longer than two months, I need to reject this submission. However, if you are able to thoroughly address the reviewers' comments, I would encourage you to resubmit.

*Reviewer #1:*

Ribosome profiling can be used to experimentally improve the annotation of genomes, identifying open reading frames that were missed by computational approaches. Recently Mankin (Mol. Cell 2019) and Storz (mBio 2019) published studies in *E. coli* in which antibiotics trap ribosomes exclusively at start sites, allowing them to identify novel ORFs inside and outside of annotated genes. Wade and co-workers employ this strategy in *M. tuberculosis*. While this study is certainly valuable in defining the proteome of this important pathogen, it does not make use of novel methods nor arrive at substantially different conclusions from these earlier studies.

Wade and co-workers claim that they have identified >1000 novel ORFs but I have some concerns about such a large number of new genes. The ribosomal profiling data (from untreated and retapamulin-treated cells) appear to be quite good. The authors show that novel ORFs on average have enriched Shine-Dalgarno sequences, lower RNA structure, and higher density at stop codons. But I want to see evidence for each novel ORF that it is translated (not the average). For ORFs outside of annotated genes, they can establish a threshold of ribosome reads that establishes confidence in translation (this data is given in table 1). Some of these values are quite low suggesting that the ORFs are not translated (at least at appreciable levels). This could also be done for the 300 or so leaderless mRNAs. For ORFs inside annotated genes, the signal at the novel start site should be compared to the signal at the annotated start site (or perhaps the ribosome density across the gene in untreated cells) and only if the novel start site signal is above some ratio should it be considered.

I also have concerns about the validation studies. A very small amount of sequence upstream of the start sites candidate genes was integrated into a different organism with a constant promoter. This means that other things that are essential for translation (transcription of that region, mRNA structure at the start site) are not taken into account. It is not surprising that start codons will initiate translation of a reporter gene if driven by a decent promoter. The question is whether or not this happens in the native genome. Validation should be done by adding an epitope tag to the 3'-end of the ORF in the native genome. I recognize that this is a lot to ask in an organism that is BSL3 with such a slow growth rate. But the strategy shown in Figure 4 doesn't work.

*Reviewer #2:*

In this manuscript, Smith and colleagues perform ribosome profiling on *M. tuberculosis* in the presence and absence of an antibiotic that blocks ribosomes at initiation. Collectively, the data support the idea that there is widespread low-level translation of small unannotated ORFs throughout the genome, most of which are unlikely to produce functional proteins. The study was rigorously and thoughtfully performed from both the experimental and analytical perspectives. The manuscript is well-written and most of the figures are clear and straightforward to interpret. The findings are significant, with a number of potentially important implications for mycobacterial physiology and evolution.

*Reviewer #3:*

The manuscript by Smith et al. describes a ribosome profiling (Ribo-seq) study of the translatome of the human pathogen *Mycobacterium tuberculosis* (Mtb). The authors map elongating and initiating (via Retapamulin treatment, which stalls initiation complexes, "ribo-RET") ribosomes on mRNAs by deep sequencing. This provides empirical evidence for translated ORFs (annotated or novel), including those within coding regions via Retapamulin-based identification of internal initiation sites. Based on genome-wide analysis of this data, the authors suggest ~1000 novel translated ORFs with high confidence and even more with lower confidence. Some of the novel ORFs, including isoforms of annotated genes or novel ORFs, are validated based on luciferase reporters or Western blots analysis of FLAG-tagged variants of the WT sequences or start-codon mutants. Finally, codon analysis suggests that most of the novel ORFs are likely non-functional as they do not show signatures of purifying selection. The study also provides a resource of novel ORFs that could be studied for functions in TB physiology. This includes small proteins, which are underrepresented in genome annotations, as well as functional studies. This is the first Ribo-seq study of Mtb, and ribo-RET has only been used in *E. coli* so far. The authors provide their data in a public web-based genome browser.

The Ribo-seq data and analysis appear solid. The major finding of the study is that there appears to be pervasive, non-productive translation in Mycobacteria, where the protein itself has no function. This mirrors the extensive antisense transcription that has also been reported, which might likewise provide substrate for evolution. Such pervasive translation has been reported previously (Meydan et al. 2019 Mol Cell, Weaver et al. MBio2019, Impens et al. Nat Microbiol. 2017). However, to much lower extend. The suggested number of >1000 novel ORFs seems rather high. While the topic is generally interesting, some additional analyses and experiments (see detailed comments) should be performed to validate the novel ORFS and strengthen the case for pervasive translation. Also, more discussion of how broadly this applies to bacteria (and/or higher organisms) and to put the study into context with previous reports of pervasive translation is also needed, in order to highlight novelty and strengthen the manuscript.

1) The introduction/discussion is rather short and appears too focused on previous work on Mycobacteria/Ribo-seq. It would be helpful to discuss also if proteomics approaches indicated pervasive translation and place the findings of this study more into context of other studies.

2) According to Figure 4C, 425 annotated ORFs were detected by ribo-RET. How many ORFs are annotated in total in Mtb and what percentage is detected? For the ones for which no TIS is found in the ribo-RET, are these not expressed under the examined conditions or why are they not captured?

3) The authors suggest that most of the novel ORFs are probably non-functional. However, they might act as regulatory elements based on translation, e.g. like attenuators or upstream ORFs Such ORFs might also be under little selection at the codon level. This should be discussed.

4) The suggested number or novel ORFs appears very high and only a small number is further validated in this study. I think it is important to provide some more evidence for them. For example, is there is any evidence from Mtb N-terminal proteomics data (Shell et al. PLOS Genet 2015) for any of these novel ORFs? This would nicely support the Ribo-seq data, even if the sensitivity allows detection of only a few.

5) Along the same lines: How many of these novel ORFs were detected in previous Ribo-seq data in M. smegmatis? Do these show hallmarks of selection at the codon level? How many are conserved in other strains? This might identify candidates for future study and strengthen the catalogue.

6) Figure 4A: What kind of start codon mutation was introduced? A stop codon? Why do the start codon mutations of the fluorescence reporters not fully abolish expression, or at least abolish expression to similar background levels for all constructs? Some of the mutations have a surprisingly small effect. Are these canonical ATG start codons? It is also unclear what is the background level (i.e. bacteria without the luciferase reporter). A positive control, i.e. an known annotated ORF and its start codon mutant, should also be included. How actively translated are these novel examples compared to annotated ORFs?

7) Figures 4B and C and S9: Some controls are missing from the Western blot validation experiments: samples from an untagged WT strain should be loaded in parallel as a control to ensure the bands are specific. Moreover, for 4C, it would be helpful to add also a start-codon mutant of Rv3709c full-length alone (to see if the smaller isoform band remains) and in combination with the start codon mutant of the isoform.

8) Figure 1 and 3: The authors show multiple times sequence coverages from ribosome profiling data. How would the sequence coverage look like for a fragmented total RNA sample as control?

9) Lines 48-50: The authors discuss library artefacts towards the 5'end. Have they tried alternative library preparation protocols to avoid that bias?

10) Can the authors show that micrococcal nuclease digestion was complete by providing gradient profiles (+/- MNase and +/- Ret). Were non-coding RNAs under-represented in the ribosome footprint libraries?

[Editors’ note: further revisions were suggested prior to acceptance, as described below.]

Thank you for submitting your article "Pervasive Translation in *Mycobacterium tuberculosis*" for consideration by *eLife*. Your article has been reviewed by 3 peer reviewers, and the evaluation has been overseen by a Reviewing Editor and Bavesh Kana as the Senior Editor. The following individual involved in review of your submission has agreed to reveal their identity: Scarlett Shell (Reviewer #1).

Reviewers felt that the current submission has been substantively improved. Some concerns remain, these primarily relate to clarifying the analysis approach. In addition, the *E. coli* comparisons emerged repeatedly as problematic amongst reviewer comments. Please reconsider how these data are presented or interpreted. Your revision should carefully address the comments by revising the text, no wet lab experiments are necessary.

1. Reviewers felt it was difficult to evaluate whether the ribosome profiling data really support so many new translated regions in Mtb, given that so much depends on what threshold is set for including or excluding candidate sites. More discussion of the threshold used and an explanation / validation in the Results section (what exactly qualifies as an IERF?) is required.

2. Are the novel genes conserved in other mycobacterial species? As the ribo-RET data do not yet exist for these species, the authors could perform bioinformatic analyses to determine whether the ORFs in question are conserved, and if so, whether there is translation in the M. smegmatis ribo-seq data already available. Conservation or lack thereof would speak to their being functional and subject to selection.

3. Reviewers concur that the *E. coli* experiments are somewhat problematic. The authors could repeat the analyses in *E. coli* and Mtb without using prior annotations. In other words, find ORFs in each genome de novo and determine how many possible start sites have associated ribo-RET density. If indeed Mtb has pervasive translation and *E. coli* does not, there should be a higher fraction in Mtb with no reference to prior annotations. This analysis would make the comparison much more compelling.

4. The authors have selected 18 novel predicted ORFs for further validation using luciferase reporters. However, it remains unclear how these ORFs were selected and if they reflect high/low confidence candidates. This is important to assess how many of the >1000 candidates (and pervasive translation as a take home message of this paper) are indeed bona fide ORFs or just artifacts. Or was the validation focused on examples that have strong signals in the sequencing data or those with strong conservation? Please clarify. In case the validated candidates just reflect the top candidates, it might be useful to shorten the list of pervasive ORFs to a higher confidence set, e.g. integrating 3-nt periodicity using smORFer (PMID: 34125903).

5. While start/stop codon patterns were revealed by the metagene analysis of the Ribo-seq data, it remains unclear how well they are reflected on a single gene basis. How many of the >1000 ORFs have this pattern? What is the "median" pattern? Or is it just approximately 100 abundant bona fide sORFs that skew the plots? Single genes with abundant read coverages in Ribo-Ret can introduce artifact peaks into metagene plots.

Other points that must be addressed:

1. Line 107-109: as a possible explanation of the displacement of the start codon peak further downstream, the authors suggest that retapamulin (RET) may not trap initiating ribosomes at leaderless mRNAs. However, it appears that the samples shown in Figure 1 were not treated with RET.

2. A leaderless mRNA should give a 15 nt ribosome footprint that would likely be missed in the study. This footprint size was confirmed by Sawyer 2021. Please show the footprint length distribution of the libraries in the supplemental information. It is puzzling why there are peaks of density 25-35 downstream of leaderless start codons. Please explain.

3. Can the Mtb ribo-seq data from Sawyer 2021 be used to validate expression of the novel ORFs described here?

4. Line 183: 1,994 IERFS. What is the threshold for the authors to call these as significant? Add one or two lines about this in the Results section. Why is this a reasonable threshold? Much depends on this seemingly arbitrary cutoff.

5. The RNA secondary structural analyses in Figure 3B – does this include all IERFs? Perhaps it should be separated in leadered vs leaderless. The region included in the RNA folding would be different, given that leaderless mRNAs have no upstream mRNA. Leaderless initiation is very sensitive to RNA structure (Bharmal, NAR Genomics and Bioinformatics, September 2021).

6. Figure 5: Perhaps there would be value in showing violin plots with the distribution of ribosome density (rpkm) and perhaps translational efficiency (ribo-seq / RNA-seq) for each gene.

7. 44 proteins expressed from novel ORFs were confirmed by MS analyses. Is there anything that these proteins have in common (biochemical similarities, high expression levels, etc) that would explain why these were detectable, but the vast majority of the new proteins were not detectable? More information about these 44 new proteins would be useful. The primary data with the y and b ion series for the three peptides confirmed by MS/MS (with synthetic standards) represent a quality control issue that could be moved to the supplementary materials.

8. Lines 76-79: Please be more specific here. It is unclear what means "large numbers" and "many sites". 10, 100, 1000?

9. Line 85: It would be helpful to indicate the genome size and total number of annotated genes at the start of this section.

10. Can the authors comment on whether they have detected any dual function sRNAs (sRNA encoding a sORF) in their data?

11. Lines 463-465: It feels that up to 10% spurious translation could be quite significant and detrimental to the cell. Can the authors comment on/clarify, what would be the cutoff for the definition of "pervasive"? 1%? 10%? 50%?

12. How are leaderless transcripts defined in this manuscript? Can leaderless also not include mRNAs with very short 5'UTRs (1-3 nt) and no Shine-Dalgarno? Does the data suggest that a 5'-end AUG/GTG etc is a requirement for leaderless initiation? Or could it also initiate 1-2nt downstream of the TSS/5'-end?

13. Please clarify what is an IERF vs an ORF? Sometimes it is not clear if the authors are requiring that an EIRF also be associated with an in-frame stop codon? Perhaps mention also that some of the detected IERFs could be due to resistance to MNase digestion?

14. Line 259: "Novel ORFs tend to be weakly expressed but efficiently translated" – this seems a bit confusing or conflicting. Would it be better to say weakly "transcribed"?

15. Lines 102-103 – "attributable to sequence biases associated with library preparation". Can the authors be more specific here? Why would library preparation biases be more prevalent at start and stop codons?

16. Lines 85-86: Simply having an RUG at the 5' position does not usually result in annotation of a leaderless gene as it also requires an in-frame stop codon. Was an RUG the only requirement in these studies? Or an RUG associated with an ORF? Please clarify.

17. Figure 3a: It would be interesting to see the proportions of different types of novel ORFs split up in the orange part of the pie chart (currently all novel ORFs are merged together).

18. Figure 7B and 7C: Please add size information on the western blots.

*Reviewer #1:*

To my reading, all of the reviewers' concerns have been thoroughly addressed. There is a substantial amount of new data validating the pervasive translation of short unannotated ORFs, and the figures have been revised for improved clarity. The quality of the data and presentation are high, and the findings are of high significance.

*Reviewer #2:*

This manuscript by Smith and co-workers uses ribosome profiling to discover new sites of translation in *Mycobacterium tuberculosis*. The data appear to be high quality, and importantly, they make use of the antibiotic retapamulin to trap newly initiated 70S ribosomes at start codons (while allowing elongating ribosomes to run-off transcripts), defining many new sites of translational initiation genome-wide. There are a stunning number of novel translated regions, more than the number of annotated genes whose translation can be detected in their experiments. The authors argue that pervasive translation occurs throughout the Mtb transcriptome.

It is difficult to evaluate whether the ribosome profiling data really support so many new translated regions in Mtb, given that so much depends on what threshold they set for including or excluding candidate sites. I would like to see more discussion of the threshold used and an explanation / validation in the Results section (what exactly qualifies as an IERF?). But I recognize that ribo-seq and retapamulin have been used in previous studies in *E. coli*, confirming the validity of the approach. And the authors argue that their novel initiation sites show enrichment of SD sequences upstream of the start codon, lower local mRNA structure, and levels of ribosome density and translational efficiency similar to known annotated genes. Furthermore, some fraction of these novel proteins can be detected by MS experiments. Together, I find these arguments compelling.

1) Are the novel genes conserved in other mycobacterial species? I realize that ribo-RET data do not yet exist for these species, but the authors could perform bioinformatic analyses to ask whether the ORFs in question are conserved, and if so, whether there is translation in the M. smegmatis ribo-seq data already available. Conservation or lack thereof would speak to their being functional and subject to selection.

2) The authors argue for pervasive translation in Mtb but not in *E. coli* on the grounds that previous ribo-RET studies in *E. coli* observed translation of a larger fraction of annotated ORFs and found fewer novel genes. This is problematic because the *E. coli* genome is arguably the best annotated genome after decades of studies. And it may be that a smaller fraction of annotated genes are expressed in Mtb under standard lab conditions than in *E. coli* under different lab conditions. Finally, it is not clear that the methods used to decide what counts as a novel start site are the same in this study and the *E. coli* study. I propose that with the *E. coli* data from Meyden 2019, the authors use exactly the same metrics for calling start sites to repeat the analyses in *E. coli* and Mtb without using prior annotations. In other words, find ORFs in each genome de novo and determine how many possible start sites have associated ribo-RET density. If indeed Mtb has pervasive translation and *E. coli* does not, there should be a higher fraction in Mtb with no reference to prior annotations. This analysis would make the comparison much more compelling.

*Reviewer #3:*

Smith et al. describe the application of ribosome profiling (Ribo-seq) and Ribo-Ret (Ribo-seq with retapamulin treatment) to *Mycobacterium tuberculosis* (Mtb) with the aim of identifying novel small ORFs (sORFs) and start codons. Ribo-seq and Ribo-Ret in Mtb surprisingly provided evidence for the translation of >1000 novel ORFs, many of which were short. In their resubmitted manuscript, the authors have carefully addressed the previous comments of the three reviewers and have added further data (e.g. additional reporter fusions and mass spectrometry-based detection of small proteins) to further support their conclusions about pervasive translation in Mycobacteria and validation on novel ORFs. Global analysis of Ribo-seq patterns at the start and stop codons, translation efficiency, as well as the newly included small protein-targeted proteomics and GC skew, supported that many of these novel ORFs are bona fide, translated ORFs. Validation of selected translation initiation regions by luciferase translational fusions and ORFs by C-terminal FLAG-tagging/western blotting support the global data. A focus is placed on leaderless ORFs, which are especially prevalent in Mtb. The authors' analysis of novel ORF conservation suggests that most of the novel ORFs are not under purifying selection, making it unclear if they have a function in Mtb. Based on this, the authors propose that Mtb experiences pervasive, apparently non-productive, translation, as has been described previously for bacterial transcriptomes.

Ribo-seq is a powerful method for monitoring translation and detection of novel ORFs. Its derivative, Ribo-Ret, has not yet been applied to many prokaryotes and is not trivial to establish or analyze. The analysis methods established and presented in this study are of interest to others applying the technique to diverse prokaryotes to overall increase confidence in Ribo-seq-predicted ORFs. Furthermore, the detected, conserved sORFs serve as a resource for the Mtb and small protein communities, and it is highly appreciated that the data has been made readily available in a browser. Compared to *E. coli*, Mtb is a high-GC organism with a unique genomic structure. The idea of pervasive translation is fairly new in prokaryotes, and the study has implications for exploring how genes arise.

Overall, this is an important study with careful analysis of the data, considerable validation, and careful response to reviewers' comments. However, I have the following suggestions to strengthen the conclusions and to further clarify certain aspects of the manuscript:

– p. 20: The authors have selected 18 novel predicted ORFs for further validation using luciferase reporters. However, it remains unclear how these ORFs were selected and if they reflect high/low confidence candidates. This is important to assess how many of the >1000 candidates (and pervasive translation as a take home message of this paper) are indeed bona fide ORFs or just artifacts? Or was the validation focused on examples that have strong signals in the sequencing data or those with strong conservation?

– In case the validated candidates just reflect the top candidates, it might be useful to shorten the list of pervasive ORFs to a higher confidence set, e.g. integrating 3-nt periodicity using smORFer (PMID: 34125903).

– While start/stop codon patterns were revealed by the metagene analysis of the Ribo-seq data, it remains unclear how well they are reflected on a single gene basis. How many of the >1000 ORFs have this pattern? What is the "median" pattern? Or is it just approximately 100 abundant bona fide sORFs that skew the plots? Single genes with abundant read coverages in Ribo-Ret can introduce artifact peaks into metagene plots.

---

## [Author Response]

[Editors’ note: the authors resubmitted a revised version of the paper for consideration. What follows is the authors’ response to the first round of review.]

Reviewer #1:Ribosome profiling can be used to experimentally improve the annotation of genomes, identifying open reading frames that were missed by computational approaches. Recently Mankin (Mol. Cell 2019) and Storz (mBio 2019) published studies in *E. coli* in which antibiotics trap ribosomes exclusively at start sites, allowing them to identify novel ORFs inside and outside of annotated genes. Wade and co-workers employ this strategy in *M. tuberculosis*. While this study is certainly valuable in defining the proteome of this important pathogen, it does not make use of novel methods nor arrive at substantially different conclusions from these earlier studies.

We agree that we have not developed any novel techniques; Ribo-RET has been applied before in *E. coli*. However, we disagree that the conclusions of our work are not substantially different to those of earlier studies”. Ribo-RET in *E. coli* identified new ORFs that we would classify as “novel” and “isoform”. However, these represented a relatively small minority of the total number of ORFs identified, as we now discuss in the manuscript. In other words, the large majority of ORFs identified by Ribo-RET in *E. coli* are annotated genes. By contrast, we detected more unannotated ORFs than annotated ORFs in *M. tuberculosis*. It is possible that our data allowed for more sensitive detection of ORFs, but this is unlikely given the enrichment of ribosome occupancy at start codons afforded by addition of retapamulin in *M. tuberculosis* versus *E. coli* (compare Figure 1B of Meydan et al., to Figure 2A in our manuscript). Hence, we conclude that pervasive translation is much more prevalent in *M. tuberculosis* than in *E. coli*.

Wade and co-workers claim that they have identified >1000 novel ORFs but I have some concerns about such a large number of new genes. The ribosomal profiling data (from untreated and retapamulin-treated cells) appear to be quite good. The authors show that novel ORFs on average have enriched Shine-Dalgarno sequences, lower RNA structure, and higher density at stop codons. But I want to see evidence for each novel ORF that it is translated (not the average). For ORFs outside of annotated genes, they can establish a threshold of ribosome reads that establishes confidence in translation (this data is given in table 1). Some of these values are quite low suggesting that the ORFs are not translated (at least at appreciable levels). This could also be done for the 300 or so leaderless mRNAs.

Like the reviewer, we were surprised to find so many novel ORFs, and we were initially skeptical of this result. Hence, we previously performed rigorous analytical and experimental follow-up work showing that (i) putative sites of translation initiation identified by Ribo-RET are far more frequently associated with likely start codons than expected by chance, (ii) novel start codons are associated with Shine-Dalgarno sequences, (iii) novel start codons are associated with reduced RNA secondary structure, (iv) novel start and stop codons have enriched ribosome density in Ribo-seq data, similar to that observed for annotated ORFs, (v) regions around novel and isoform start codons drive translation in a luciferase reporter construct, and (vi) two selected novel ORFs and two selected isoform ORFs are supported by western blot data. As the reviewer notes, these data do not assign confidence scores to individual ORFs, but rather demonstrate a pattern of pervasive translation. We now provide additional support for individual novel ORFs in the form of mass spectrometry data.

To further address the reviewer’s concern, we have determined the ribosome density across novel ORF transcripts by normalizing Ribo-seq coverage to RNA-seq coverage, a common measure of relative ribosome occupancy. Normalizing to RNA-seq data accounts for the large variability in RNA levels for the set of novel ORFs; thus, low Ribo-seq signal could reflect efficient translation of an RNA that is of low abundance. We limited our analysis to the regions of novel ORFs that do not overlap an annotated ORF on the same strand, including an additional 30 nt at either end of annotated ORFs to account for the footprints of initiating/terminating ribosomes that extend beyond the ORF boundaries. It is important to note that the novel ORFs analyzed are likely a biased set, enriched for ORFs located antisense to annotated genes; however, avoiding overlap with annotated genes is essential to interpret these data. For the remaining 871 novel ORFs, some represented in full and others represented only partially, RNA-seq and Ribo-seq coverage tends to be lower than for annotated ORFs, but ribosome density per transcript is only slightly reduced. Moreover, ribosome density per transcript tends to be much higher for novel ORFs than for a set of control non-coding transcripts selected based on transcription start site data for regions not close to ORFs. Thus, our data clearly show that most of the novel ORF transcripts are robustly translated, even if many are weakly expressed. These analyses are included as a new figure and supplementary figure (Figure 5, and Figure 5, Figure Supplement 1). We have also added normalized ribosome density measurements for individual ORFs to Supplementary Tables 1 and 3; we anticipate that readers will use these numbers to assign confidence to individual ORF calls.

For ORFs inside annotated genes, the signal at the novel start site should be compared to the signal at the annotated start site (or perhaps the ribosome density across the gene in untreated cells) and only if the novel start site signal is above some ratio should it be considered.

It is impossible to assign Ribo-seq or RNA-seq signal to a single ORF if it overlaps another ORF. Hence, we have chosen to limit the analysis described above to novel ORFs that do not overlap annotated genes.

I also have concerns about the validation studies. A very small amount of sequence upstream of the start sites candidate genes was integrated into a different organism with a constant promoter. This means that other things that are essential for translation (transcription of that region, mRNA structure at the start site) are not taken into account. It is not surprising that start codons will initiate translation of a reporter gene if driven by a decent promoter. The question is whether or not this happens in the native genome. Validation should be done by adding an epitope tag to the 3'-end of the ORF in the native genome. I recognize that this is a lot to ask in an organism that is BSL3 with such a slow growth rate. But the strategy shown in Figure 4 doesn't work.

Our interpretation of the nLuc reporter fusion data is that the selected sequences *can* function as ribosome binding sites; we have added text to note that the sequences are not in their native context.

We agree that the optimal version of the western blot experiment would be to introduce epitope tags at the native locus in *M. tuberculosis*. However, this is technically challenging and would take ~6 months. Hence, we opted to introduce the tagged ORFs into *M. smegmatis*, including the complete predicted 5’ UTR in each case. We used a strong promoter to rule out the possibility that an *M. tuberculosis* promoter might be transcriptionally inactive in *M. smegmatis*. We disagree that any potential start codon would be efficiently translated if it were located within a well-transcribed RNA; our Ribo-RET and Ribo-seq data clearly show that this is not the case. We also note that for the isoform ORFs, the overlapping ORF serves as an internal control for the western blot that accounts for transcript levels. Nonetheless, we agree with the reviewer’s assertion that many sequences could function as start codons if present in an RNA, and indeed we believe this is the basis for the pervasive translation we observe.

To address the reviewer’s concern, we have used mass spectrometry to identify proteins translated from the novel ORFs. While there are major technical challenges for identification of small proteins by mass spectrometry, as we now discuss in the manuscript, we detected 44 proteins translated from novel ORFs.

Reviewer #3:The manuscript by Smith et al. describes a ribosome profiling (Ribo-seq) study of the translatome of the human pathogen Mycobacterium tuberculosis (Mtb). The authors map elongating and initiating (via Retapamulin treatment, which stalls initiation complexes, "ribo-RET") ribosomes on mRNAs by deep sequencing. This provides empirical evidence for translated ORFs (annotated or novel), including those within coding regions via Retapamulin-based identification of internal initiation sites. Based on genome-wide analysis of this data, the authors suggest ~1000 novel translated ORFs with high confidence and even more with lower confidence. Some of the novel ORFs, including isoforms of annotated genes or novel ORFs, are validated based on luciferase reporters or Western blots analysis of FLAG-tagged variants of the WT sequences or start-codon mutants. Finally, codon analysis suggests that most of the novel ORFs are likely non-functional as they do not show signatures of purifying selection. The study also provides a resource of novel ORFs that could be studied for functions in TB physiology. This includes small proteins, which are underrepresented in genome annotations, as well as functional studies. This is the first Ribo-seq study of Mtb, and ribo-RET has only been used in *E. coli* so far. The authors provide their data in a public web-based genome browser.The Ribo-seq data and analysis appear solid. The major finding of the study is that there appears to be pervasive, non-productive translation in Mycobacteria, where the protein itself has no function. This mirrors the extensive antisense transcription that has also been reported, which might likewise provide substrate for evolution. Such pervasive translation has been reported previously (Meydan et al. 2019 Mol Cell, Weaver et al. MBio2019, Impens et al. Nat Microbiol. 2017). However, to much lower extend. The suggested number of >1000 novel ORFs seems rather high. While the topic is generally interesting, some additional analyses and experiments (see detailed comments) should be performed to validate the novel ORFS and strengthen the case for pervasive translation. Also, more discussion of how broadly this applies to bacteria (and/or higher organisms) and to put the study into context with previous reports of pervasive translation is also needed, in order to highlight novelty and strengthen the manuscript.

We have added new controls and a new mass spectrometry experiment that strengthen the case for pervasive translation in *M. tuberculosis*.

1) The introduction/discussion is rather short and appears too focused on previous work on Mycobacteria/Ribo-seq. It would be helpful to discuss also if proteomics approaches indicated pervasive translation and place the findings of this study more into context of other studies.

We have greatly expanded the Discussion, and we have added text comparing our work to other related studies. In the new section of the Results that describes proteomic data, we discuss the limitations of standard techniques that greatly limit the number of small proteins that can be detected. For these reasons, we have chosen not to compare our work to proteomic studies.

2) According to Figure 4C, 425 annotated ORFs were detected by ribo-RET. How many ORFs are annotated in total in Mtb and what percentage is detected? For the ones for which no TIS is found in the ribo-RET, are these not expressed under the examined conditions or why are they not captured?

See response to Reviewer #1.

3) The authors suggest that most of the novel ORFs are probably non-functional. However, they might act as regulatory elements based on translation, e.g. like attenuators or upstream ORFs Such ORFs might also be under little selection at the codon level. This should be discussed.

We have added a section on this topic in the Discussion. uORFs might have atypical codon usage because they are embedded within an RNA structure that is conserved because they are involved in regulation, and/or because rare codons participate in regulation. To investigate this question, we have now looked at codon usage in a set of six previously described regulatory uORFs that are enriched in cysteine codons. Only one of the individual uORFs has significant G/C skew (Fisher’s exact test *p* < 0.05), and the group of ORFs as a whole does not have significant G/C skew; this is despite strong evidence that these are conserved, regulatory uORFs. We repeated the analysis after removing the cysteine codons, reasoning that they are likely essential for the uORF regulatory functions, and they are expected to reduce G/C skew. Indeed, removing the cysteine codons increased the G/C skew for all ORFs, and the G/C skew for the set of uORFs as a group was much higher, and statistically significant (Fisher’s exact test *p* < 0.001). We now discuss the implications of this analysis for the full set of novel ORFs.

4) The suggested number or novel ORFs appears very high and only a small number is further validated in this study. I think it is important to provide some more evidence for them. For example, is there is any evidence from Mtb N-terminal proteomics data (Shell et al. PLOS Genet 2015) for any of these novel ORFs? This would nicely support the Ribo-seq data, even if the sensitivity allows detection of only a few.

See comments in response to Reviewer 1. In short, we believe the Ribo-seq data provide very strong evidence that the majority of novel ORFs are robustly translated. Nonetheless, we have added mass spectrometry data as suggested, providing independent experimental support for 44 novel ORFs. We note that the small size of most of the novel ORFs makes them difficult to detect using standard proteomic approaches, which are biased strongly against small proteins. We attempted to enrich for small proteins, but this clearly requires further optimization. We have also determined the ribosome density per transcript for many of the novel ORFs; these data further support the idea that the novel ORFs are efficiently translated.

5) Along the same lines: How many of these novel ORFs were detected in previous Ribo-seq data in M. smegmatis? Do these show hallmarks of selection at the codon level? How many are conserved in other strains? This might identify candidates for future study and strengthen the catalogue.

To address the question of whether novel ORFs are functional, we have ranked non-overlapping novel ORFs by their G/C-skew, highlighting ORFs that are likely under purifying selection. Thus, we identify ~90 likely functional novel ORFs. Comparison to data from *M. smegmatis* is confounded by the difficulty in identifying novel ORFs from standard Ribo-seq data. We plan to address this in a future manuscript that includes Ribo-RET data for *M. smegmatis*.

6) Figure 4A: What kind of start codon mutation was introduced? A stop codon? Why do the start codon mutations of the fluorescence reporters not fully abolish expression, or at least abolish expression to similar background levels for all constructs? Some of the mutations have a surprisingly small effect. Are these canonical ATG start codons? It is also unclear what is the background level (i.e. bacteria without the luciferase reporter). A positive control, i.e. an known annotated ORF and its start codon mutant, should also be included. How actively translated are these novel examples compared to annotated ORFs?

The start codons were mutated to RCG. The luciferase assay data are plotted on a log scale, so the effect of mutating the start codon in each case is large. Nonetheless, most of the mutant constructs retain some activity. Interestingly, we observed the same phenomenon when investigating novel ORFs in *E. coli*, where we integrated the luciferase reporter at the native chromosomal locus (https://www.biorxiv.org/content/10.1101/2021.07.02.450978v2). We speculate that there are alternative internal start codons, or that non-canonical start codon sequences can be used, as described for *E. coli* (PMID 28334756). As a control, we have included sequences for three annotated ORF; these behave similarly to the other reporters, giving us confidence in our conclusion that the novel/isoform RBSs are functional.

7) Figures 4B and C and S9: Some controls are missing from the Western blot validation experiments: samples from an untagged WT strain should be loaded in parallel as a control to ensure the bands are specific. Moreover, for 4C, it would be helpful to add also a start-codon mutant of Rv3709c full-length alone (to see if the smaller isoform band remains) and in combination with the start codon mutant of the isoform.

The start codon mutants serve as a control, and the blots looking at different proteins, all of which use the same epitope tag, control for each other when detecting non-specific bands. For isoform ORFs, we did not mutate the start codon of the full-length gene because this would likely lead to premature Rho-dependent transcription termination upstream of the isoform start codon.

8) Figure 1 and 3: The authors show multiple times sequence coverages from ribosome profiling data. How would the sequence coverage look like for a fragmented total RNA sample as control?

We have added this control dataset, which does not show enrichment 15/12 nt downstream of start/stop codons (Figure 4, Figure Supplement 1B-D). We do observe the artifactual signal precisely at start/stop codons that we attribute to imprecise sequence read alignment due to these libraries being made by polyadenylation of RNA fragments. We have also included a control analysis of mock ORFs (Figure 4, Figure Supplement 1A). Specifically, we identified mock ORFs whose stop codons do not match any annotated ORF or any ORF we identified by Ribo-RET. We limited the set of mock ORFs to those found in regions that are detectably transcribed. We do not observe Ribo-seq signal enrichment 15/12 nt downstream of the start/stop codons of the mock ORFs.

9) Lines 48-50: The authors discuss library artefacts towards the 5'end. Have they tried alternative library preparation protocols to avoid that bias?

We have not, but our data suggest that this would be a good idea for any groups applying Ribo-seq methods to bacteria.

10) Can the authors show that micrococcal nuclease digestion was complete by providing gradient profiles (+/- MNase and +/- Ret). Were non-coding RNAs under-represented in the ribosome footprint libraries?

We do not have gradient profiles for the samples used for Ribo-RET and Ribo-seq. However, the clear enrichment of ribosome occupancy at start and stop codons shows that MNase processing was sufficient. Non-coding RNAs are strongly under-represented in the Ribo-seq data relative to RNA-seq, as now indicated in Figure 5 and Figure 5, Figure Supplement 1 (new figures).

[Editors’ note: what follows is the authors’ response to the second round of review.]

Essential revisions:Reviewers felt that the current submission has been substantively improved. Some concerns remain, these primarily relate to clarifying the analysis approach. In addition, the *E. coli* comparisons emerged repeatedly as problematic amongst reviewer comments. Please reconsider how these data are presented or interpreted. Your revision should carefully address the comments by revising the text, no wet lab experiments are necessary.1. Reviewers felt it was difficult to evaluate whether the ribosome profiling data really support so many new translated regions in Mtb, given that so much depends on what threshold is set for including or excluding candidate sites. More discussion of the threshold used and an explanation / validation in the Results section (what exactly qualifies as an IERF?) is required.

We have added the details for IERF identification to the relevant part of the Results. The most important parameter in identifying IERFs is the minimum read coverage in the Ribo-RET data. The value we chose (5.5 reads per million) is arbitrary; a higher value would identify fewer IERFs and hence fewer ORFs, but with a lower FDR. We tried to strike a balance between detecting more translated ORFs and keeping the FDR low. We do not believe that there is a defined set of translated ORFs; rather, there is a continuum of expression levels for all possible ORFs, such that a lower threshold of discovery or a more sensitive approach will identify more translated ORFs.

2. Are the novel genes conserved in other mycobacterial species? As the ribo-RET data do not yet exist for these species, the authors could perform bioinformatic analyses to determine whether the ORFs in question are conserved, and if so, whether there is translation in the M. smegmatis ribo-seq data already available. Conservation or lack thereof would speak to their being functional and subject to selection.

This is an interesting and important question, but we believe it is beyond the scope of the current study. tBLASTn analysis reveals potential homologues for >100 non-overlapping, novel ORFs, with only about a third of these having significant G/C-skew. However, a more sophisticated analysis is required to determine whether conservation is due to selective pressure on the ORF itself or overlapping sequence features. This is something we are working on for a future publication. Similarly, we have generated Ribo-RET data for *M. smegmatis* to identify novel ORFs. Comparison of the *Mtb* and *M. smegmatis* datasets will be the subject of a future paper.

3. Reviewers concur that the E. coli experiments are somewhat problematic. The authors could repeat the analyses in E. coli and Mtb without using prior annotations. In other words, find ORFs in each genome de novo and determine how many possible start sites have associated ribo-RET density. If indeed Mtb has pervasive translation and *E. coli* does not, there should be a higher fraction in Mtb with no reference to prior annotations. This analysis would make the comparison much more compelling.

We have chosen to remove the text suggesting that *Mtb* has a larger proportion of translation dedicated to novel ORFs than *E. coli*. While our data suggest that this is the case, there are enough technical differences between the studies of the two species that we do not feel we can make this claim with sufficient confidence, even if we were to analyze the data using the same pipeline.

4a. The authors have selected 18 novel predicted ORFs for further validation using luciferase reporters. However, it remains unclear how these ORFs were selected and if they reflect high/low confidence candidates. This is important to assess how many of the >1000 candidates (and pervasive translation as a take home message of this paper) are indeed bona fide ORFs or just artifacts. Or was the validation focused on examples that have strong signals in the sequencing data or those with strong conservation? Please clarify.

They are all from the top 25^th^ percentile of Ribo-RET scores, but they cover a broad range of values for ribosome density per transcript (median percentile rank of 37 for the 8 ORFs that could be assessed in non-overlapping regions). We have added these details to the text in the Results section. We believe the MS data provide a much better assessment of the novel ORFs identified by Ribo-RET. Hence, we have moved the luciferase assay data and the western blot data to the supplement.

4b. In case the validated candidates just reflect the top candidates, it might be useful to shorten the list of pervasive ORFs to a higher confidence set, e.g. integrating 3-nt periodicity using smORFer (PMID: 34125903).

There is strong evidence from our data and from published *E. coli* Ribo-seq data (PMID 27924019) that the 3 nt periodicity in bacterial Ribo-seq data is due to sequence biases within codons (e.g. G/C-skew), and does not reflect the codon-by-codon movement of ribosomes across ORFs.

5. While start/stop codon patterns were revealed by the metagene analysis of the Ribo-seq data, it remains unclear how well they are reflected on a single gene basis. How many of the >1000 ORFs have this pattern? What is the "median" pattern? Or is it just approximately 100 abundant bona fide sORFs that skew the plots? Single genes with abundant read coverages in Ribo-Ret can introduce artifact peaks into metagene plots.

For each ORF that contributes data to the metagene plots, values are normalized to the maximum value in the analyzed region. Thus, all ORFs are contribute equally; no single ORF can dominate.

Other points that must be addressed:1. Line 107-109: as a possible explanation of the displacement of the start codon peak further downstream, the authors suggest that retapamulin (RET) may not trap initiating ribosomes at leaderless mRNAs. However, it appears that the samples shown in Figure 1 were not treated with RET.

This sentence refers to Ribo-seq data without drug treatment. We observe an enrichment of signal 15 nt downstream of leadered ORF start codons even without RET treatment, albeit to a lesser degree (e.g. Figure 1A).

2. A leaderless mRNA should give a 15 nt ribosome footprint that would likely be missed in the study. This footprint size was confirmed by Sawyer 2021. Please show the footprint length distribution of the libraries in the supplemental information. It is puzzling why there are peaks of density 25-35 downstream of leaderless start codons. Please explain.

We have chosen not to make a plot of the distribution of sequence read lengths because it is the mappable sequence reads that count, and the standard read-mapping tools ignore very short reads. We note that Sawyer *et al.* showed similar sequence read coverage around leaderless start codons as our data. We agree that the peak of ribosome footprint 3’ density 25-30 nt downstream of leaderless start codons is an interesting observation, but it is one that we cannot explain. We presume it reflects the fundamentally different mechanism of leaderless translation initiation.

3. Can the Mtb ribo-seq data from Sawyer 2021 be used to validate expression of the novel ORFs described here?

Ribo-seq data from the Sawyer *et al.* study have substantially lower enrichment of signal downstream of start and stop codons, likely because cells were treated with chloramphenicol before harvesting. Hence, these data are not as useful for validating the novel ORF calls.

4. Line 183: 1,994 IERFS. What is the threshold for the authors to call these as significant? Add one or two lines about this in the Results section. Why is this a reasonable threshold? Much depends on this seemingly arbitrary cutoff.

See response to major comment #1.

5. The RNA secondary structural analyses in Figure 3B – does this include all IERFs? Perhaps it should be separated in leadered vs leaderless. The region included in the RNA folding would be different, given that leaderless mRNAs have no upstream mRNA. Leaderless initiation is very sensitive to RNA structure (Bharmal, NAR Genomics and Bioinformatics, September 2021).

This analysis only considers leadered ORFs. Since the leaderless ORFs were inferred based on transcription start site data, there is no expectation that the secondary structure at the start of these RNAs will be lower than that of random sequence.

6. Figure 5: Perhaps there would be value in showing violin plots with the distribution of ribosome density (rpkm) and perhaps translational efficiency (ribo-seq / RNA-seq) for each gene.

We have added supplementary figure panels showing cumulative frequency plots for relative RNA coverage and relative ribosome density (Figure 5 —figure supplement 1B-C). We think Figure 5B is a good way to show the ribosome density-per-transcript numbers.

7. 44 proteins expressed from novel ORFs were confirmed by MS analyses. Is there anything that these proteins have in common (biochemical similarities, high expression levels, etc) that would explain why these were detectable, but the vast majority of the new proteins were not detectable? More information about these 44 new proteins would be useful. The primary data with the y and b ion series for the three peptides confirmed by MS/MS (with synthetic standards) represent a quality control issue that could be moved to the supplementary materials.

The y and b ion series data are important to show that the peptides we detected from cell lysates were correctly assigned to their corresponding proteins. We did not identify any features strongly enriched in the proteins we validated by MS, although the proteins we detected by MS tend to be associated with ORFs with higher ribosome occupancy per transcript than ORFs associated with MS-undetected proteins (Mann-Whitney U Test *p* = 0.03). Note that the MS-validated proteins are indicated in Supplementary Tables 1 and 3, so readers can see the various features associated with these proteins.

8. Lines 76-79: Please be more specific here. It is unclear what means "large numbers" and "many sites". 10, 100, 1000?

We have replaced references to “large numbers” with specific ranges. We have kept “many sites” because we do not believe the number of pervasively translated ORFs can be quantified. In other words, we don’t believe there is a discrete set of pervasively translated ORFs.

9. Line 85: It would be helpful to indicate the genome size and total number of annotated genes at the start of this section.

Done.

10. Can the authors comment on whether they have detected any dual function sRNAs (sRNA encoding a sORF) in their data?

There are annotated sRNAs for which we observe Ribo-seq coverage consistent with translation of an sORF, but these will be the subject of a future paper.

11. Lines 463-465: It feels that up to 10% spurious translation could be quite significant and detrimental to the cell. Can the authors comment on/clarify, what would be the cutoff for the definition of "pervasive"? 1%? 10%? 50%?

We are making the point here that our data suggest that pervasive translation contributes less than 10% of all translation, although it could be considerably less than that. By our definition, there is no cut-off for the amount of translation required to be defined as “pervasive”, although we acknowledge that the physiological relevance of pervasive translation will depend to a large extent on the extent of pervasive translation.

12. How are leaderless transcripts defined in this manuscript? Can leaderless also not include mRNAs with very short 5'UTRs (1-3 nt) and no Shine-Dalgarno? Does the data suggest that a 5'-end AUG/GTG etc is a requirement for leaderless initiation? Or could it also initiate 1-2nt downstream of the TSS/5'-end?

We define leaderless transcripts as having no 5’ UTR (stated on line 43). Recent data from the Schrader lab indicate that very short 5’ UTRs can be tolerated in *Caulobater crescentus*. That is likely to also be the case in mycobacteria, but we have not investigated this in detail. The effect of very short 5’ UTRs on leaderless translation in mycobacteria will be the subject of a future paper.

13. Please clarify what is an IERF vs an ORF? Sometimes it is not clear if the authors are requiring that an EIRF also be associated with an in-frame stop codon? Perhaps mention also that some of the detected IERFs could be due to resistance to MNase digestion?

As discussed in more detail in our response to major comment #1, we have added text to the Results to clarify how IERFs were identified. In brief, an IERF is a site of enrichment in the Ribo-RET data. Most IERFs correspond to a site of translation initiation, but ~30% were not assigned to an ORF, likely for a variety of reasons, e.g. the ORF uses a non-canonical start codon, the spacing between the start codon and the enriched Ribo-RET signal does not fall within the range we required, the Ribo-RET signal is not due to ribosome occupancy but rather to association with another large complex, the Ribo-RET signal is due to ribosomes stalled for a reason other than being trapped at a start codon. Without more data, we are reluctant to comment in the manuscript on specific reasons why some IERFs are not associated with expected start codons at expected positions.

14. Line 259: "Novel ORFs tend to be weakly expressed but efficiently translated" – this seems a bit confusing or conflicting. Would it be better to say weakly "transcribed"?

We have changed “expressed” to “transcribed”, as suggested. We previously used “expressed” because we wanted to include the possibility that RNA stability contributes to the RNA level, but we are comfortable saying transcription.

15. Lines 102-103 – "attributable to sequence biases associated with library preparation". Can the authors be more specific here? Why would library preparation biases be more prevalent at start and stop codons?

The MNase bias is the same everywhere, but start codons always have a “T” at the second position, and stop codons always have a “T” at the first position, so the bias lines up across all ORFs. The same effect would be observed if metagene plots were made for any specific nucleotide sequence, e.g. all AAT codons. We have added some text to clarify this point: “We note that there are also smaller peaks and troughs of Ribo-seq signal precisely at start and stop codons, likely attributable to sequence biases associated with library preparation that are highlighted when similar sequences (e.g. start/stop codons) are aligned”.

16. Lines 85-86: Simply having an RUG at the 5' position does not usually result in annotation of a leaderless gene as it also requires an in-frame stop codon. Was an RUG the only requirement in these studies? Or an RUG associated with an ORF? Please clarify.

We are not sure we understand the question. Every RUG will have an in-frame stop somewhere downstream. The 1,285 transcripts referred to are those that begin with RUG. If a 5’ RUG is sufficient for leaderless translation, the locations of leaderless ORFs can be inferred from the positions of the transcription start sites.

17. Figure 3a: It would be interesting to see the proportions of different types of novel ORFs split up in the orange part of the pie chart (currently all novel ORFs are merged together).

We have expanded the figure to indicate the different subclasses of novel and isoform ORF. We also now indicate the different subclasses of novel ORF in Supplementary Tables 1 and 3.

18. Figure 7B and 7C: Please add size information on the western blots.

Done.